# REINFORCEMENT LEARNING FOR SADDLE-POINT EQUILIBRIA WITHOUT FULL STATE EXPLORATION

## ABSTRACT

We introduce a new fixed-point condition on the state-action-value $Q$-function for zero-sum Markov turn games that suffices to construct saddle-point and security policies, but is less restrictive than the classical condition arising from the Bellman equation. We then propose an iterative algorithm that guarantees convergence to a function satisfying this less restrictive condition. The key benefit of the new condition and algorithm is that convergence to a saddle-point can (and typically will) be reached without full exploration of the state-space; generally enabling the solution of larger games with less computation. Our algorithm is based on a limited form of exploration that gathers samples from repeated attempts to certify the current candidate policies as a saddle-point, motivating the terminology "saddle-point exploration" (SPE). We illustrate the use of the new condition/algorithms in several combinatorial games that can be scaled in terms of the size of the state and action spaces. Numerical results, using both tabular and neural network $Q$-function representations, consistently show that saddle-point policies can be formally certified without full state exploration and, for several games, we can see that the fraction of states explored *decreases* as the size of the game grows.

## 1 INTRODUCTION

We address two-player zero-sum Markov turn games with finite but large state spaces, for which the goal is to find minimax policies with "modest" computation. In this context, *minimax* or *security* policies refer to policies $\pi_1^*, \pi_2^*$ that achieve the outer maxima in the following worst-case optimizations

$$\max_{\pi_1 \in \Pi_1} \min_{\pi_2 \in \Pi_2} J_1(\pi_1, \pi_2), \qquad\qquad \max_{\pi_2 \in \Pi_2} \min_{\pi_1 \in \Pi_1} J_2(\pi_1, \pi_2), \qquad (1)$$

where $\Pi_i$, $i \in \{1, 2\}$ is the policy space for player $\mathsf{P}_i$ and $J_i(\pi_1, \pi_2)$ this player's expected sum of future rewards, with $J_1(\pi_1, \pi_2) = -J_2(\pi_1, \pi_2)$. We use the qualifier "modest" to mean that we seek to certify policies to be solutions to (1) without exploring the full state-space of the game. The terminology *turn games* means that only one player is allowed to make a decision at each state (Sidford et al., 2019; Jia et al., 2019; Shah et al., 2020; Anderson et al., 2025). Turn games generalize *alternate play games* in which players always alternate in making decisions, like chess, checkers, or go. As opposed to general zero-sum Markov games (e.g., Filar & Vrieze (1997)), turn games with finite state and action spaces have pure saddle-point policies under very mild assumptions, avoiding the need to consider mixed or behavioral policies (Hespanha, 2017).

$Q$-learning, which was originally developed by Watkins (1989) for single-player Markov decision processes and later extended to two-player zero-sum games by Littman (1994); Littman & Szepesvári (1996), remains the most widely used provably correct approach to construct minimax policies. $Q$-learning performs an iterative computation of the state-action-value function, generally called the $Q$-function, that assigns to each state-action pair the associated (minimax) future rewards. Correctness of this approach relies on the observation that the iteration converges to a unique fixed point, which is the optimal $Q$-function. In practice, the $Q$-learning iteration typically terminates by either explicitly checking whether or not the fixed-point condition holds (often up to some prescribed acceptable error) or by using a fixed number of iterations for which one can guarantee convergence (up to some prescribed acceptable error). Both options require evaluating the $Q$-function over the whole state-space; either explicitly in the first case, or implicitly in the second case, because the available

sample complexity bounds that guarantee convergence of the $Q$-function rely on full state-exploration (Even-Dar & Mansour, 2003; Hu & Wellman, 2003; Beck & Srikant, 2012; Wainwright, 2019; Chen et al., 2020; Zhang et al., 2020; Ménard et al., 2021; Li et al., 2021; Lee, 2023; Li et al., 2024).

Our first contribution is a new condition on a candidate $Q$-function that suffices to guarantee that the policies extracted from it are a solution to (1). This condition, which we call "restricted fixed point," is expressed as a fixed-point equality on a restricted subset of the state space and can be checked without full state exploration. While the usual (unrestricted) fixed-point condition typically only has a unique solution — precisely the optimal $Q$-function — our restricted condition typically has multiple solutions, many of which are *not* the optimal $Q$-function. Regardless, we show in Theorem 2 that all functions satisfying the restricted condition lead to minimax policies in the sense of (1).

The second contribution is an algorithm that guarantees convergence to a restricted fixed point (Algorithm 1). This algorithm relies on updates to the $Q$-function that are similar to the classical updates (e.g., by Littman & Szepesvári (1996)) adapted to turn games; but it differs from previous work in two aspects: termination condition and sample selection/exploration. Termination is based on checking a saddle-point condition that involves solving two "inner-loop" optimization problems using single-player $Q$-learning. The proposed algorithm is named "saddle-point exploration" (SPE) because, beyond a termination condition, these inner-loop optimizations provide all the samples that are needed for the (outer-loop) $Q$-function updates. Embedding two inner-loop $Q$-learning iterations within an outer-loop iteration might seem to result in a very inefficient algorithm. However, this is not the case, because the outer loop makes use of all the samples generated during the inner-loop optimizations, which is enabled by the off-policy $Q$-learning updates. We prove that Algorithm 1 terminates in finite time for deterministic turn games (Theorem 3), while for stochastic turn games we prove an analogous high-probability result (Theorem 4). In both cases, visiting the entire reachable state space is not required for termination (although in the worst case it is unavoidable).

SPE works with general $Q$-function representations of the candidate saddle-point, including tabular and neural network forms. For the latter representation, instead of asking for convergence of the neural network during training — typically a difficult condition to verify — the algorithm only requires that the policies derived from the neural network satisfy the saddle-point conditions. The complexity of this verification is relatively low, because it presumes that the policy of one of the players is frozen, greatly reducing the reachable state-space.

We illustrate the benefits of the SPE algorithm by applying it to a collection of scalable board games available in the OpenSpiel software package (Lanctot et al., 2019), including Hex, Y, Breakthrough, Clobber, Dots and Boxes; as well as the strategy game Atlatl (Rood, 2022; Darken, 2025). For all these games, we observe that SPE terminates without full reachable state exploration. Moreover, by considering multiple versions of the same game with different board sizes and time-horizon, we observe that the fraction of reachable states explored before termination either stabilizes to some percentage as the size of the game increases, or actually decreases.

## RELATED WORK

In recent years, significant work has been devoted towards the *sample complexity analysis of Q-learning*; specifically on determining a minimum number of samples for which the policies arising from the $Q$-learning iteration can be certified as optimal (Even-Dar & Mansour, 2003; Beck & Srikant, 2012; Wainwright, 2019; Chen et al., 2020). For one-player problems, fairly sharp sample complexity bounds can be found in (Li et al., 2024), where it is shown that the number of samples required for (synchronous) $Q$-learning to obtain an $\epsilon$-accurate estimate of the "exact" $Q$-function scales with $|\mathcal{S}| \times |\mathcal{A}| \times H^4/\epsilon^2$ (up to logarithmic factors), where $\mathcal{S}$ and $\mathcal{A}$ denotes the state and action spaces, respectively, and $H \coloneqq 1/(1-\gamma)$ is an "effective" time horizon for a $\gamma \in (0,1)$-discounted infinite-horizon cost. This result is "sharp" in $H$ and $\epsilon$ in the sense that the authors provide an MDP for which the $Q$-function requires a number of order $H^4/\epsilon^2$ to converge. Results of this nature for zero-sum Markov games include an additional term $|\mathcal{B}|$ for the opponent's action space (Lee, 2023).

It has previously been recognized that it is possible to construct almost-optimal policies from samples without convergence of the $Q$-function. *Regret-based analyses* accomplish this by bounding the number of samples required to obtain a policy with a small cumulative cost/reward difference compared to the optimal policy. One of the tightest results under this setup was reported by Li et al. (2021), who show accumulated regret over $N$ episodes in a finite-horizon setting with episode length

$T$ of order $\sqrt{|\mathcal{S}| \times |\mathcal{A}| \times T^2 \times N}$, but only after $N \geqslant |\mathcal{S}| \times |\mathcal{A}| \times T^{10}$. While different works obtain different scaling laws for the regret, the minimum sample size, and the memory complexity, the existing regret-based analyses of $Q$-learning still require a sample size on the order of $|\mathcal{S}| \times |\mathcal{A}| \times T$ or greater (Li et al., 2021; Zhang et al., 2020; Ménard et al., 2021).

Additional results that are specific for *two-player zero-sum Markov games* include (Bai et al., 2020), which provides a variant of Nash $Q$-learning (Hu & Wellman, 2003) with sample complexity bounds to achieve an $\epsilon$-approximate Nash-equilibrium on the order of $|\mathcal{S}| \times |\mathcal{A}| \times |\mathcal{B}| \times T^5/\epsilon^2$. More recently, Feng et al. (2024) reduce the polynomial dependence on the horizon to $T^3$ and obtain the minimax-optimal dependence on $T$, $|\mathcal{S}|$ and $\epsilon$. Shreyas & Vijesh (2024) proposed a multi-step approach that converges with probability one in the setting with discounted rewards.

It should be noted that, while our restricted saddle-point condition can be used to certify a policy as optimal with computational complexity below $|\mathcal{S}| \times |\mathcal{A}|$, it is possible to construct games for which the only restricted saddle point is the usual (unrestricted saddle point) and no benefits can be gained.

SPE's test of the saddle-point condition can be viewed as trying to find weaknesses in the current candidate policies, which has similarities with the "Golf with Exploiter" algorithm in Jin et al. (2022). In that work, iterations are performed over a set of state-value functions from which an optimistic policy is extracted, as well as the best response against it. A probabilistic guarantee of convergence is provided in terms of the Bellman Eluder dimension of the game, which for Markov games with a tabular representation is upper-bounded by the size of the state-action space, but can be smaller. The key challenge with this work lies in devising algorithms that efficiently iterate over a set of value functions, which is defined by a growing number of constraints posed on these sets by the samples. While we arrived at the SPE algorithm from a very different approach (based on the restricted fixed-point condition), the SPE algorithm can be viewed as a practical implementation of some of the ideas in (Jin et al., 2022).

In addition to the references above, there is a large body of work on *developing heuristic algorithms to solve large zero-sum turn games*: these include AlphaZero (Silver et al., 2017), AlphaStar (Vinyals et al., 2019) which uses a variant of Policy Space Response Oracles (Lanctot et al., 2017), and Monte Carlo Tree Search methods (Silver et al., 2016). However, the focus of these algorithms has not been on termination with correctness guarantees.

## 2 ZERO-SUM TURN MARKOV GAMES

We consider *Markov games* with state $s_t$ at time $t \geqslant 0$, taking value in a state-space $\mathcal{S}$. In *turn games*, only one player can make a decision at each state, so the state-space $\mathcal{S}$ can be partitioned into two disjoint sets $\mathcal{S}_1, \mathcal{S}_2$ with the understanding that, when $s_t$ belongs to $\mathcal{S}_1$, the action $a_t \in \mathcal{A}$ is selected by player $\mathsf{P}_1$. Otherwise, $s_t \in \mathcal{S}_2$ and the action is selected by player $\mathsf{P}_2$. To simplify the notation, we use the same symbol $\mathcal{A}$ to denote the set of actions available to both players, with the understanding that when $s_t \in \mathcal{S}_i$, the elements of $\mathcal{A}$ should be viewed as the options available to $\mathsf{P}_i$, $i \in \{1, 2\}$.

In *zero-sum games*, the rewards for the two players add up to zero and we denote by $r_{t+1} \in \mathcal{R} \subset \mathbb{R}$, $t \geqslant 0$ the immediate reward collected by the player that selected the action $a_t$ at time $t$. The total reward collected by player $\mathsf{P}_i$, $i \in \{1, 2\}$ for the initial state $s_0 \in \mathcal{S}$ is then given by

$$J_i(s_0) := \sum_{t=0}^{\infty} \mathrm{E}[r_{t+1}\, \mathrm{sgn}_i(s_t)], \tag{2}$$

where $\mathrm{sgn}_i(s_t) = 1$ if $s_t \in \mathcal{S}_i$ and $\mathrm{sgn}_i(s_t) = -1$ otherwise. The sets $\mathcal{S}, \mathcal{A}, \mathcal{R}$ are assumed finite and the state $s_t$ is a *stationary controlled Markov chain* in the sense that

$$\mathrm{P}(s_{t+1} = s', r_{t+1} = r \mid s_t = s, a_t = a) = p(s', r \mid s, a) \tag{3}$$

$\forall t \geqslant 0,\ s, s' \in \mathcal{S},\ a \in \mathcal{A},\ r \in \mathcal{R}$; where $p : \mathcal{S} \times \mathcal{R} \times \mathcal{S} \times \mathcal{A} \to [0, 1]$ is the *transition/reward probability function*. We say that a game is *deterministic* if $p(\cdot, \cdot)$ only takes values in the set $\{0, 1\}$ and that the game *terminates in finite time* if there exists a finite time $T \geqslant 1$ such that $r_t = 0$, $\forall t \geqslant T$ with probability one, regardless of the actions $a_t \in \mathcal{A}$ selected. Games with finite horizon can be trivially reduced to games with infinite horizon but finite termination time, by incorporating time into the state and creating an absorbing "game-over" state with zero-reward to which the state is forced to transition once the end of the time horizon is reached.

## 2.1 POLICIES AND VALUE FUNCTION FOR TURN GAMES

A policy for the player $\mathsf{P}_i$, $i \in \{1, 2\}$ is a deterministic map $\pi_i : \mathcal{S}_i \to \mathcal{A}$ that selects the action $a_t = \pi_i(s_t)$ when the state $s_t$ is in $\mathcal{S}_i$. The finite set of all such deterministic policies is denoted by $\Pi_i$. We recall that, for turn games, there is no advantage in considering stochastic policies (Anderson et al., 2025). For a pair of policies $(\pi_1, \pi_2) \in \Pi_1 \times \Pi_2$, we define the *policy pair's value function* as

$$V_{\pi_1, \pi_2}(s) := \operatorname{sgn}_i(s) \sum_{t=\tau}^{\infty} \mathrm{E}_{\pi_1, \pi_2}[r_{t+1} \operatorname{sgn}_i(s_t) \mid s_\tau = s] \quad \forall s \in \mathcal{S}, i \in \{1, 2\}, \tag{4}$$

where the subscripts in $\mathrm{E}_{\pi_1, \pi_2}[\cdot]$ highlight that the expectation is conditioned to the actions determined by the given policies. We get the same value for $i = 1$ and $i = 2$ because $\operatorname{sgn}_1(s) = -\operatorname{sgn}_2(s)$, $\forall s \in \mathcal{S}$. The time $\tau \geqslant 0$ from which the summation is started does not affect its value due to the stationarity of the Markov chain. It is straightforward to verify that the reward (2) collected by player $\mathsf{P}_i$ can be obtained from the value function using

$$J_i(s_0) = \operatorname{sgn}_i(s_0) V_{\pi_1, \pi_2}(s_0), \quad \forall i \in \{1, 2\}. \tag{5}$$

## 2.2 SADDLE-POINTS AND SECURITY POLICIES

A pair of policies $(\pi_1^*, \pi_2^*)$ is a *(pure) $\epsilon$-saddle-point* for some $\epsilon \geqslant 0$, if

$$J_1^*(s_0) := \operatorname{sgn}_1(s_0) V_{\pi_1^*, \pi_2^*}(s_0) \geqslant \max_{\pi_1 \in \Pi_1} \operatorname{sgn}_1(s_0) V_{\pi_1, \pi_2^*}(s_0) - \epsilon, \tag{6a}$$

$$J_2^*(s_0) := \operatorname{sgn}_2(s_0) V_{\pi_1^*, \pi_2^*}(s_0) \geqslant \max_{\pi_2 \in \Pi_2} \operatorname{sgn}_2(s_0) V_{\pi_1^*, \pi_2}(s_0) - \epsilon, \tag{6b}$$

and, for $\epsilon = 0$, $J_1^*(s_0) = -J_2^*(s_0)$ is called the *value of the game*. When omitted, $\epsilon = 0$ is assumed. In view of (5), the equality in (6a) with $\epsilon = 0$ expresses no regret in the sense that $\mathsf{P}_1$ does not regret its choice of $\pi_1^*$ (over any other policy $\pi_1$) against $\pi_2^*$ and, similarly, (6b) with $\epsilon = 0$ expresses no regret for $\mathsf{P}_2$. Saddle-point policies are known to also be *security policies with values $J_1^*(s_0)$ and $J_2^*(s_0)$ for players $\mathsf{P}_1$ and $\mathsf{P}_2$, respectively*; in the sense that

$$J_1^*(s_0) = \max_{\pi_1 \in \Pi_1} \min_{\pi_2 \in \Pi_2} \operatorname{sgn}_1(s_0) V_{\pi_1, \pi_2}(s_0) = \min_{\pi_2 \in \Pi_2} \operatorname{sgn}_1(s_0) V_{\pi_1^*, \pi_2}(s_0) \tag{7a}$$

$$J_2^*(s_0) = \max_{\pi_2 \in \Pi_2} \min_{\pi_1 \in \Pi_1} \operatorname{sgn}_2(s_0) V_{\pi_1, \pi_2}(s_0) = \min_{\pi_1 \in \Pi_1} \operatorname{sgn}_2(s_0) V_{\pi_1, \pi_2^*}(s_0) \tag{7b}$$

which means that, by using the policy $\pi_i^*$, the player $\mathsf{P}_i$ can expect a reward at least as large as $J_i^*(s_0)$, *no matter what policy the other player uses* (see, e.g., (Hespanha, 2017)).

Assuming that the per-state reward is bounded, games with infinite horizon but discounted costs, can always be "truncated" to a game with a sufficiently large but finite termination time $T$, so that the costs of the truncated and the original games differ by less than some arbitrarily small value $\eta > 0$, regardless of the policies used by the players. In this case, using the results in this paper to compute an $\epsilon$-saddle-point for the truncated game, automatically gives us a $(\epsilon + 2\eta)$-saddle-point to the original infinite-horizon discounted game.

## 2.3 FIXED-POINT SUFFICIENT CONDITION FOR SADDLE-POINT

Saddle-point and security policies can be easily constructed provided that we can find a function $Q : \mathcal{S} \times \mathcal{A} \to \mathbb{R}$ that is a *fixed point* of

$$Q(s, a) = \mathrm{E}\left[r_{t+1} + \operatorname{sgn}_1(s_t) \operatorname{sgn}_1(s_{t+1}) \max_{a' \in \mathcal{A}} Q(s_{t+1}, a') \mid s_t = s, a_t = a\right], \tag{8}$$

$\forall s \in \mathcal{S}$, $a \in \mathcal{A}$. The terminology "fixed point" arises from regarding the right-hand side of (8) as the action of an operator that acts on $Q$, and produces the same function $Q$. The following result provides an explicit formula (10) for saddle-point policies as a function of the fixed point $Q$. To express it, we need the following definition: we say that a function $V : \mathcal{S} \to \mathbb{R}$ is *absolutely summable* if for every pair of policies $(\pi_1, \pi_2) \in \Pi_1 \times \Pi_2$ for players $\mathsf{P}_1, \mathsf{P}_2$, respectively, the series

$$\sum_{t=\tau}^{\infty} \mathrm{E}_{\pi_1, \pi_2}[V(s_t) \mid s_\tau = s], \tag{9}$$

is absolutely convergent for every $s \in \mathcal{S}, \tau \geqslant 0$. For games that terminate in finite time $T$, any function $V : \mathcal{S} \to \mathbb{R}$ for which $V(s_t) = 0$, $\forall t \geqslant T$ with probability one is absolutely summable since the series degenerates into a finite summation.

**Theorem 1** (Fixed-point sufficient condition). *Suppose there exists a function $Q^* : \mathcal{S} \times \mathcal{A} \to \mathbb{R}$ that is a fixed point of* (8) *and $V^*(s) \coloneqq \max_{a \in \mathcal{A}} Q^*(s, a)$, $\forall s \in \mathcal{S}$ is absolutely summable. Then any pair of policies $(\pi_1^*, \pi_2^*)$ for which*

$$\pi_i^*(s) \in \arg \max_{a \in \mathcal{A}} Q^*(s, a), \quad \forall s \in \mathcal{S}_i, \ i \in \{1, 2\} \tag{10}$$

*is a saddle-point and these are policies, with values $J_1^*(s_0) = \mathrm{sgn}_1(s_0) V^*(s_0) = -J_2^*(s_0)$.* $\quad\square$

We state this result without proof since it can be derived from classical results (Littman, 1994; Littman & Szepesvári, 1996), at least when the operator defined by the right-hand side of (8) is a strict contraction. It also follows from a more general result to be derived shortly.

*Q-learning* can be used to iteratively construct a function $Q : \mathcal{S} \times \mathcal{A} \to \mathbb{R}$ that satisfies the fixed-point condition in (8). In the context of zero-sum turn games, *Q-learning* starts from some initial estimate $Q^0 : \mathcal{S} \times \mathcal{A} \to \mathbb{R}$ and iteratively draws samples $(s_t, a_t, s_{t+1}, r_{t+1})$ from the transition/reward probability function $p(s_{t+1}, r_{t+1} \mid s_t, a_t)$, each leading to an update of the form

$$Q^{k+1}(s_t, a_t) = (1 - \alpha_k) Q^k(s_t, a_t) + \alpha_k Q_{\text{target}}^{k+1}, \tag{11}$$

for some sequence $\alpha_k \in (0, 1]$ and $Q_{\text{target}}^{k+1} \coloneqq r_{t+1} + \mathrm{sgn}_1(s_t) \mathrm{sgn}_1(s_{t+1}) \max_{a' \in \mathcal{A}} Q^k(s_{t+1}, a')$. Under mild assumptions on the operator defined by the right-hand side of (8) and the sequence $\alpha_k$, this iteration converges to the unique fixed point of (8) when every element of $\mathcal{S} \times \mathcal{A}$ appears infinitely many times in the sample sequence $\{(s_t, a_t)\}$ (Tsitsiklis, 1994).

## 3 RESTRICTED FIXED POINT

To define "restricted fixed point" we need the following definitions: given a set or pairs of policies $\Pi \subset \Pi_1 \times \Pi_2$, we define the *set $\mathcal{S}_\Pi$ of reachable states under $\Pi$* to contain all states that can be reached with positive probability under such policies, i.e.,

$$\mathcal{S}_\Pi \coloneqq \big\{ s \in \mathcal{S} : \exists t \geqslant 0, \ (\pi_1, \pi_2) \in \Pi \text{ such that } \mathrm{P}_{\pi_1, \pi_2}(s_t = s) > 0 \big\}. \tag{12}$$

We say that a function $Q^\dagger : \mathcal{S} \times \mathcal{A} \to \mathbb{R}$ is a *restricted fixed point of* (8) when this equation holds over $(s, a) \in \mathcal{S}_{\Pi^\dagger} \times \mathcal{A}$, where $\mathcal{S}_{\Pi^\dagger}$ is the set of reachable states under

$$\Pi^\dagger \coloneqq \big\{ (\pi_1^\dagger, \pi_2) : \pi_2 \in \Pi_2 \big\} \cup \big\{ (\pi_1, \pi_2^\dagger) : \pi_1 \in \Pi_1 \big\}, \tag{13}$$

where

$$\pi_i^\dagger(s) \in \arg \max_{a \in \mathcal{A}} Q^\dagger(s, a), \quad \forall s \in \mathcal{S}_i, \ i \in \{1, 2\}. \tag{14}$$

Every fixed point (over the whole $(s, a) \in \mathcal{S} \times \mathcal{A}$) is necessarily a restricted fixed point, but the converse is not true since the set $\mathcal{S}_{\Pi^\dagger}$ is typically much smaller than the whole state-space $\mathcal{S}$. Moreover, fixed points are often unique (e.g., when the right-hand side of (8) defines the action of an operator that is a strict contraction), but restricted fixed points are generally not unique. Nevertheless, restricted fixed points still enable the construction of saddle-point and security policies:

**Theorem 2** (Restricted fixed-point sufficient condition). *Suppose there exists a function $Q^\dagger$ that is a restricted fixed point of* (8) *and for which $V^\dagger(s) \coloneqq \max_{a \in \mathcal{A}} Q^\dagger(s, a)$, $\forall s \in \mathcal{S}$ is absolutely summable. Then the pair $(\pi_1^\dagger, \pi_2^\dagger)$ is a saddle-point and these are security policies, with values $J_1^*(s_0) = \mathrm{sgn}_1(s_0) V^\dagger(s_0) = -J_2^*(s_0)$.* $\quad\square$

The proof of Theorem 2 is included in Appendix A.1. This proof directly shows that the restricted fixed-point condition suffices to establish that the saddle-point conditions in (6) hold and takes advantage of the observation that (6) only involves value functions $V_{\pi_1, \pi_2}$ for $(\pi_1, \pi_2) \in \Pi^\dagger$. However, this derivation cannot use the relationship between the $Q^\dagger$ and dynamic-programming's cost-to-go, since (8) will generally not hold over $\mathcal{S}$ for restricted fixed points.

## 4 SADDLE-POINT EXPLORATION (SPE) ALGORITHM

We now describe an algorithm that, like in classical $Q$-learning, constructs an iterative sequence of functions $Q^k$ that are updated according to (11) and from which we will construct saddle-point

policies. However, unlike in classical $Q$-learning, our goal now is to update $Q^k$ to get convergence to a *restricted fixed point* rather than a regular fixed point. To accomplish this, the SPE Algorithm 1 selects the samples for (11) by using the current iterate $Q^k$ to construct a candidate saddle-point

$$\pi_1^k(s) \in \arg\max_{a\in\mathcal{A}} Q^k(s,a), \ \forall s \in \mathcal{S}_1, \qquad \pi_2^k(s) \in \arg\max_{a\in\mathcal{A}} Q^k(s,a), \ \forall s \in \mathcal{S}_2, \qquad (15)$$

and checks whether these policies form a saddle-point, which justifies the terminology *saddle-point exploration*. Specifically, the code in lines 7–12 fixes $\mathsf{P}_1$'s policy at $\pi_1^k$ and uses (single-player) $Q$-learning to find $\mathsf{P}_2$'s best-response policy $\pi_2^{\mathrm{br}}$. The sequence of functions $\{Q_2^0, \ldots, Q_2^{k_2}\}$ is used for this purpose and, upon convergence, the best response has the form

$$\pi_2^{\mathrm{br}}(s) \in \arg\max_{a\in\mathcal{A}} Q_2^{k_2}(s,a), \quad \forall s \in \mathcal{S}_2, \qquad (16)$$

and its use against $\pi_1^k$ results in a reward for $\mathsf{P}_2$ equal to

$$\mathrm{sgn}_2(s_0) V_{\pi_1^k, \pi_2^{\mathrm{br}}}(s_0) = \mathrm{sgn}_2(s_0) \max_{a\in\mathcal{A}} Q_2^{k_2}(s_0). \qquad (17)$$

Similarly, the code in lines 15–20 computes $\mathsf{P}_1$'s best-response policy $\pi_1^{\mathrm{br}}$ to $\pi_2^k$, which has the form

$$\pi_1^{\mathrm{br}}(s) \in \arg\max_{a\in\mathcal{A}} Q_1^{k_1}(s,a), \quad \forall s \in \mathcal{S}_1, \qquad (18)$$

and its use against $\pi_2^k$ results in rewards for $\mathsf{P}_1$ equal to

$$\mathrm{sgn}_1(s_0) V_{\pi_1^{\mathrm{br}}, \pi_2^k}(s_0) = \mathrm{sgn}_1(s_0) \max_{a\in\mathcal{A}} Q_1^{k_1}(s_0). \qquad (19)$$

The termination condition in line 22 essentially guarantees that $\pi_1^k$ and $\pi_2^k$ satisfy a saddle-point condition like (7).

We emphasize that the convergence of the sequences $\{Q_1^{k_1}\}$ and $\{Q_2^{k_2}\}$ only requires exploration of the subsets of the state-space that are reachable when either $\mathsf{P}_1$'s policy is frozen at $\pi_1^k$ or when $\mathsf{P}_2$'s policy is frozen at $\pi_2^k$. Even though this test may need to be performed several times, we shall see that the SPE algorithm typically terminates without selecting a single sample from a large subset of the reachable states in $\mathcal{S}$. Nevertheless, the samples gathered still suffice to guarantee convergence of $Q^k$ to a restricted fixed point. In view of this, line 4 could be skipped altogether, but we include it because additional samples provide opportunities to speed up termination (see Appendix A.5).

## 4.1 DETERMINISTIC GAMES

While SPE is applicable to stochastic zero-sum turn games (not necessarily with finite state and action spaces, or termination time), the remainder of this section is focused on finite deterministic games, for which it is straightforward to be precise on two items that were left open: how to represent the functions $Q^k, Q_1^{k_2}, Q_2^{k_1}$, and how to check in lines 12, 20 that $Q_1^{k_2}, Q_2^{k_1}$ have converged.

For deterministic games, the learning rate in (11) can be set to $\alpha_k = 1, \forall k$, which makes establishing convergence of $Q_1^{k_1}$ relatively simple: it suffices to keep track of the last iteration number at which the update (11) resulted in a change in $Q_1^{k_1}$ and ensuring, since then, (i) every state $s_t \in \mathcal{S}$ and every action $a_t \in \mathcal{A}$ that can be reached when $\mathsf{P}_1$'s policy is fixed at $\pi_1^k$ appeared at least once in the samples generated in line 9, and (ii) none of these updates led to an actual change in $Q_1^{k_1}$. Convergence of $Q_2^{k_2}$ can be similarly tested. We assumed here that we can perform "exact" updates for $Q_1^{k_1}$ and $Q_2^{k_2}$, which typically requires a tabular representation. While this may seem restrictive for large games, it is important to recall that these functions presume that the (deterministic) policy of one of the players has been fixed, which typically greatly reduces the size of the reachable state-space. In fact, our numerical experiments show that, when using hash tables of the board configuration to represent $Q_1^{k_1}$ and $Q_2^{k_2}$, the loops in 7–12 and 15–20 converge quickly and the tables remain small.

Convergence of the sequence $Q^k$ does not need to be checked because the exit condition in line 22 only involves $Q_1^{k_1}$ and $Q_2^{k_2}$. This provides much greater flexibility in representing $Q^k$, which can be represented by a deep neural network trained through a batch update using the samples collected in lines 9 and 17. However, the theoretical results that follow assume an exact update for $Q^k$.

The following assumption is needed to establish the correctness for deterministic games.

---

**Algorithm 1** $Q$-learning with saddle-point exploration (SPE)

1: initialize $Q^0(s, a) = 0, \forall s \in \mathcal{S}, a \in \mathcal{A}$ and set $k \leftarrow 0$
2: **loop**
3:   ▷ *(Optional) exploration*                                                    ◁
4:   generate any number of samples $\{(s_t, a_t, s_{t+1}, r_{t+1})\}$ from (3) using any algorithm
5:   extract P$_1$'s policy $\pi_1^k$ from $Q^k$ using (15)
6:   ▷ *proceed by computing* P$_2$*'s best response against* $\pi_1^k$                ◁
7:   initialize $Q_2^0(s, a) = 0, \forall s \in \mathcal{S}, a \in \mathcal{A}$ and set $k_2 \leftarrow 0$
8:   **repeat**
9:     generate sample(s) $(s_t, a_t, s_{t+1}, r_{t+1})$ from (3), restricting $a_t = \pi_1^k(s_t)$ when $s_t \in \mathcal{S}_1$
10:    use sample(s) to update $Q_2^{k_2+1}$ using (11)
11:    $k_2 \leftarrow k_2 + 1$
12:  **until** the function $Q_2^{k_2}(\cdot, \cdot)$ has converged          (see Sections 4.1, 4.2)
13:  extract P$_2$'s policy $\pi_2^k$ from $Q^k$ using (15)
14:  ▷ *proceed by computing* P$_1$*'s best response against* $\pi_2^k$                ◁
15:  initialize $Q_1^0(s, a) = 0, \forall s \in \mathcal{S}, a \in \mathcal{A}$ and set $k_1 \leftarrow 0$
16:  **repeat**
17:    generate sample(s) $(s_t, a_t, s_{t+1}, r_{t+1})$ from (3), restricting $a_t = \pi_2^k(s_t)$ when $s_t \in \mathcal{S}_2$
18:    use sample(s) to update $Q_1^{k_1+1}$ using (11)
19:    $k_1 \leftarrow k_1 + 1$
20:  **until** the function $Q_1^{k_1}(\cdot, \cdot)$ has converged          (see Sections 4.1, 4.2)
21:  ▷ *termination condition*                                                    ◁
22:  **if** $\operatorname{sgn}_2(s_0) \max_{a \in \mathcal{A}} Q_2^{k_2}(s_0, a) + \operatorname{sgn}_1(s_0) \max_{a \in \mathcal{A}} Q_1^{k_1}(s_0, a) \leqslant \eta$ **then** terminate
23:  ▷ *update* $Q^k$ *using samples collected above*                                 ◁
24:  **for all** samples $(s_t, a_t, s_{t+1}, r_{t+1})$ collected in lines 4, 9, and 17 **do**
25:    update $Q^{k+1}$ using (11)
26:    $k \leftarrow k + 1$

---

**Assumption 1** (Exploration). The algorithms use to generate the sequences of samples in lines 9 and 17 guarantee that

1. If the iteration in lines 7-12 did not converge, every pair $(s_t, a_t)$ in $\mathcal{S}_{\Pi_2^k} \times \mathcal{A}$ would appear infinitely often in the sequence of samples in line 9, where $\mathcal{S}_{\Pi_2^k}$ denotes the set of states reachable under $\Pi_2^k := \{(\pi_1, \pi_2^k) : \pi_1 \in \Pi_1\}$.

2. If the iteration in lines 15-20 did not converge, every pair $(s_t, a_t)$ in $\mathcal{S}_{\Pi_1^k} \times \mathcal{A}$ would appear infinitely often in the sequence of samples in line 17, where $\mathcal{S}_{\Pi_1^k}$ denotes the set of states reachable under $\Pi_1^k := \{(\pi_1^k, \pi_2) : \pi_2 \in \Pi_2\}$.

**Theorem 3** (SPE Algorithm 1 for deterministic games). *Assume that the state and action spaces are finite and that the game is deterministic, terminates in finite time, all updates of $Q$, $Q_1$, $Q_2$ use $\alpha_k = 1$, $\eta \geqslant 0$, and Assumption 1 holds. At every iteration, the following bounds on the security values hold:*

$$\operatorname{sgn}_1(s_0) \max_{a \in \mathcal{A}} Q_2^{k_2}(s_0, a) \leqslant \max_{\pi_1 \in \Pi_1} \min_{\pi_2 \in \Pi_2} \operatorname{sgn}_1(s_0) V_{\pi_1, \pi_2}(s_0) \leqslant \operatorname{sgn}_1(s_0) \max_{a \in \mathcal{A}} Q_1^{k_1}(s_0, a) \quad (20a)$$

$$\operatorname{sgn}_2(s_0) \max_{a \in \mathcal{A}} Q_1^{k_1}(s_0, a) \leqslant \max_{\pi_2 \in \Pi_2} \min_{\pi_1 \in \Pi_1} \operatorname{sgn}_2(s_0) V_{\pi_1, \pi_2}(s_0) \leqslant \operatorname{sgn}_2(s_0) \max_{a \in \mathcal{A}} Q_2^{k_2}(s_0, a) \quad (20b)$$

*and the difference between the upper and lower bounds becomes no larger than $\eta$ when the algorithm terminates. Moreover, the algorithm terminates after a finite number of iterations and, upon termination at iteration $k$, the pair $(\pi_1^k, \pi_2^k)$ is an $\eta$-saddle-point. For $\eta = 0$, $J_1^*(s_0) = \operatorname{sgn}_1(s_0) \max_{a \in \mathcal{A}} Q_1^{k_1}(s_0, a) = -J_2^*(s_0)$ is the value of the game.* □

The proof of Theorem 3 is included in Appendix A.3, but the basic arguments proceeds as follows: We start by showing that $P_1$'s policy $\pi_1^{\mathrm{br}}$ (18) is optimal against $P_2$'s policy $\pi_2^k$ with the rewards in (19) and that $P_2$'s policy $\pi_2^{\mathrm{br}}$ (16) is optimal against $P_1$'s policy $\pi_1^k$, with the rewards in (17). Once this has been established, we show that the termination condition guarantees that the pair $(\pi_1^k, \pi_2^k)$ satisfies the saddle-point conditions (6). The proof that the algorithm terminates in finite time then relies on showing that $Q^k$ converges in a finite number of steps to a restricted fixed point of (8).

## 4.2 STOCHASTIC GAMES

For stochastic games, we cannot expect exact convergence of $Q_1^{k_1}$ and $Q_2^{k_2}$ in lines 7–12 and 15–20 to take place in finite time. Instead, we assume that the number of samples is sufficiently large to guarantee that, upon exit of these loops, the functions $Q_1^{k_1}$ and $Q_2^{k_2}$ are no more than $\epsilon > 0$ away from the "optimal" with high probability. Specifically, $Q_2^{k_2}$ reflects the value of $P_2$'s best response policy against $P_1$'s policy $\pi_1^k$ with an error smaller than $\epsilon$; whereas $Q_1^{k_1}$ reflects the value of $P_1$'s best response against $P_2$'s policy $\pi_2^k$ with an error also smaller than $\epsilon$:

**Assumption 2** (Convergence in stochastic setting). There exists a constant $\epsilon \geqslant 0$ and a sequence $\{\delta_\ell \in [0,1) : \ell \geqslant 1, \sum_{\ell=1}^{\infty} \delta_\ell \leqslant \delta^\dagger < 1\}$ such that, for every $\ell \geqslant 1$, the $\ell$-th time that the loops in lines 7–12 and 15–20 are executed, the number of samples is sufficiently large, so as to guarantee that

$$| \operatorname{sgn}_2(s_0) \max_{a \in \mathcal{A}} Q_2^{k_2}(s_0) - \max_{\pi_2 \in \Pi_2} \operatorname{sgn}_2(s_0) V_{\pi_1^k, \pi_2}(s_0)| \leqslant \epsilon, \tag{21}$$

$$| \operatorname{sgn}_1(s_0) \max_{a \in \mathcal{A}} Q_1^{k_1}(s_0) - \max_{\pi_1 \in \Pi_1} \operatorname{sgn}_1(s_0) V_{\pi_1, \pi_2^k}(s_0)| \leqslant \epsilon. \tag{22}$$

with probability at least $1 - \delta_\ell$, with the "failure probability" $\delta_\ell$ independent across tests. $\qquad \square$

Bounds on the number of samples required for Assumption 2 to hold can be found in several of the references provided in Section 1. Making sure that $\sum_{\ell=1}^{\infty} \delta_\ell \leqslant \delta^\dagger$ can be accomplished, e.g., with $\delta_\ell = O(1/\ell^2)$, which typically requires the number of samples used in lines 7–12 and 15–20 to increase as the number of tests $\ell$ increases. However, the dependence of the number of samples on $\delta_\ell$ is typically logarithmic (Li et al., 2024) so $\delta_\ell = O(1/\ell^2)$ leads to a mild (logarithmic in $\ell$) increase in the number of samples per test. In addition to Assumption 2, we need to make sure that repeated executions of the loops in lines 7–12 and 15–20, generate a diverse sets of samples:

**Assumption 3** (Cross-loop independence). If the loops in lines 7–12 and 15–20 are called infinitely times with $(\pi_1^k, \pi_2^k)$ equal to the same pair $(\pi_1^\dagger, \pi_2^\dagger) \in \Pi_1 \times \Pi_2$, then

$$\sum_{t=0}^{\infty} \alpha_k I_{s,a}(s_{t_k}, a_{t_k}) \overset{\mathrm{wpo}}{=} +\infty, \qquad \sum_{t=0}^{\infty} \alpha_k^2 I_{s,a}(s_{t_k}, a_{t_k}) \overset{\mathrm{wpo}}{\leqslant} C < \infty, \qquad \forall s \in \mathcal{S}_{\Pi^\dagger}, a \in \mathcal{A}$$

for some finite constant $C$, where $\mathcal{S}_{\Pi^\dagger}$ is defined by (12)–(14) and

$$I_{s,a}(\bar{s}, \bar{a}) = \begin{cases} 1 & \bar{s} = s, \ \bar{a} = a \\ 0 & \text{otherwise.} \end{cases} \qquad \square$$

This type of assumption is common in stochastic approximation arguments, except that here it does not need to hold over the whole state-action space $\mathcal{S} \times \mathcal{A}$. Instead, it only needs to hold over sets $\mathcal{S}_{\Pi^\dagger}$, generated by policies that arise infinitely many times in the sequence $(\pi_1^k, \pi_2^k)$.

**Theorem 4** (SPE Algorithm 1 for stochastic games). *Assume that the state and action spaces are finite, the game terminates in finite time, $\eta > 2\epsilon$, and Assumptions 2, 3 hold. After the $\ell$th test, the following bounds on the security values hold:*

$$\operatorname{sgn}_1(s_0) \max_{a \in \mathcal{A}} Q_2^{k_2}(s_0, a) - \epsilon \leqslant \max_{\pi_1 \in \Pi_1} \min_{\pi_2 \in \Pi_2} \operatorname{sgn}_1(s_0) V_{\pi_1, \pi_2}(s_0) \leqslant \operatorname{sgn}_1(s_0) \max_{a \in \mathcal{A}} Q_1^{k_1}(s_0, a) + \epsilon \tag{23a}$$

$$\operatorname{sgn}_2(s_0) \max_{a \in \mathcal{A}} Q_1^{k_1}(s_0, a) - \epsilon \leqslant \max_{\pi_2 \in \Pi_2} \min_{\pi_1 \in \Pi_1} \operatorname{sgn}_2(s_0) V_{\pi_1, \pi_2}(s_0) \leqslant \operatorname{sgn}_2(s_0) \max_{a \in \mathcal{A}} Q_2^{k_2}(s_0, a) + \epsilon \tag{23b}$$

*with probability at least $1 - \delta_\ell$. Moreover, the algorithm terminates after a finite number of iterations with probability 1 and, upon termination at iteration $k$, the pair $(\pi_1^k, \pi_2^k)$ is an $(2\epsilon + \eta)$-saddle-point with probability at least $1 - \delta^\dagger$.* $\qquad \square$

The proof of Theorem 4 follows similar steps to that of Theorem 3, but now uses a stochastic approximation argument two show that $Q^k$ must eventually converge to a restricted fixed point of (8). The condition $\sum_{\ell=1}^{\infty} \delta_\ell \leqslant \delta^\dagger$ essentially guarantees that the probability of error does not accumulate too rapidly across multiple tests of the exit condition.

## 5 NUMERICAL RESULTS

To demonstrate the performance of Algorithm 1, we use several games available in OpenSpiel (Lanctot et al., 2019) and the strategy game Atlatl (Rood, 2022; Darken, 2025). These games were chosen because for all we can select the "board size" and for some we can also select the duration of the game. This enables us to create game families with varying state-space sizes, where the games within each family can be meaningfully compared to each other in terms of state-space coverage.

### 5.1 BASELINE

We use for term of comparison a lower bound on the number of iterations required by *every* algorithm whose correctness is based on convergence of the $Q$-function over the entire $\mathcal{S} \times \mathcal{A}$. As discussed in Section 1, all such algorithms (as well as all the regret-based algorithms, also discussed there) can only guarantee correctness if the number of samples exceeds $|\mathcal{S}| \times \mathrm{Poly}(T)$, where $\mathrm{Poly}(T) \geqslant 1$ represents a polynomial function of the time horizon $T$. For simplicity, we are ignoring the multiplicative factor $|\mathcal{A}|$ that is the same for all algorithms. In reality, $\mathcal{S}$ only needs to contain states that can be reached from the game's initial state $s_0$, so we use for lower bound the size $|\mathcal{S}_{\Pi_1 \times \Pi_2}|$ of the reachable states. For fairness, we use the (very optimistic) lower bound $\mathrm{Poly}(T) \geqslant 1$, because our algorithm essentially uses a replay buffer for the update of $Q^k$ in lines 24–26, which allows us to reuse the same sample multiple times with little additional computational cost (Mnih et al., 2013). Alternative off-policy algorithms that use large replay buffers can similarly decrease the number of game samples to one per state, potentially reducing the required number of game samples to $|\mathcal{S}_{\Pi_1 \times \Pi_2}|$.

In the results below, we compute $|\mathcal{S}_{\Pi_1 \times \Pi_2}|$ using an exhaustive search across the state-space. Even though we use a fairly efficient search algorithm to determine $|\mathcal{S}_{\Pi_1 \times \Pi_2}|$, for some of largest games this exhaustive search does not terminate within a reasonable compute time limit (24h) and therefore we are not able to provide a comparison.

### 5.2 ALGORITHM 1 IMPLEMENTATION DETAILS

We consider two representations for the $Q$-function in (15): (i) a tabular representation hashed by a bit-vector embedding of the game board; and (ii) a Deep $Q$-Network (DQN) whose input is the same bit-vector state embedding with one output per action, as in (Mnih et al., 2013). For the functions $Q_1, Q_2$ used in the termination checks in lines 7–12 and 15–20, we only used the tabular representation because, as noted in Section 4, this greatly facilitates checking for convergence and it can be very efficient even for relatively large games. To satisfy Assumption 1, we do exploration using tempered Boltzmann policies (Anderson et al., 2025), but the results would not change significantly if we used the more common $\epsilon$-greedy exploration. We present a more extensive set of results for option (i) above, but include a few examples for option (ii) to demonstrate that SPE also works with other $Q$-function representations.

All results were obtained with a Julia implementation. We call Atlatl using a Julia wrapper to its Python interface. We use a 2021 M1 Max chip with 10 cores and 32GB RAM. For all the experiments we only include results that can be solved in less than 24 hours of run time.

### 5.3 RESULTS

We first show that Algorithm 1 is able to find a saddle-point without sampling large portions of the reachable state space. We consider several square board sizes for Hex, Y, Breakthrough, and Clobber, and run Algorithm 1 for each of them. We numerically verify we have obtained a saddle-point with associated security policies by solving the optimizations in lines 7-12. In Figure 1(left), we plot the number of states that were explored by Algorithm 1 as a fraction of our baseline $|\mathcal{S}_{\Pi_1 \times \Pi_2}|$ for both the tabular (purple) and DQN (red) representations. The games in this figure have up to about 8 million reachable states. For tabular representations (purple), we can see that the fraction of states

explored decreases as the size of the game increases (with the exception of $5 \times 5$ Y). We present fewer results for the the DQN representation (red), but the results appear to be comparable to the tabular representations.

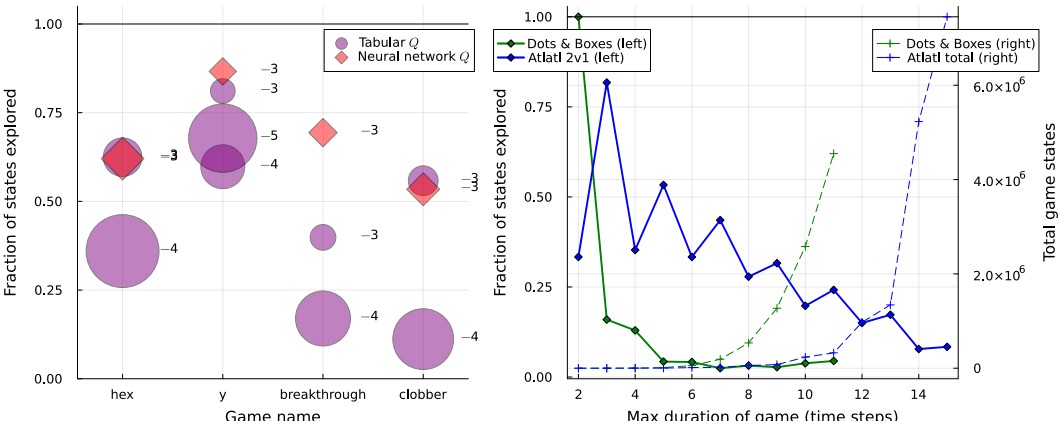

Figure 1: (left) We run Algorithm 1 for square board games of size $n \times n$ (labeled $-n$) and show (left) the fraction of reachable states explored. The size of each marker is proportional to the logarithm of the total number of reachable states, with the $4 \times 4$ Hex game containing about 8 million states. (right) The left-axis shows the fractional exploration versus the duration of the game for both Atlatl and Dots and Boxes; the right-axis of the plot indicates exponential growth in the states as the game duration increases.

We then examine how the results scale with the duration of the game (number of moves played) for Dots and Boxes on a $3 \times 3$ board and for the "2v1" Atlatl scenario; as both games are still meaningful over a variable time horizon. We observe in Figure 1(right) that, as the duration increases, the fraction of states explored by Algorithm 1 decreases exponentially for Atlatl. For Dots and Boxes the fraction decreases significantly up until games with 7 moves and then roughly stabilizes but — when compared to the growth rate of the total number of reachable states (on the right-axis) — this indicates the total number of states explored by Algorithm 1 grows at a more favorable rate. In fact, for Dots and Boxes Algorithm 1 can solve this game up to its maximum duration of 24 moves in less than 8 hours and exploring less than 10M states, whereas we were not able to do an exhaustive exploration of the reachable state space for more than 11 moves in 24 hours. For the "2v1" Atlatl scenario, Algorithm 1 can solve a game with 20 time steps in less than 18 hours, exploring a little less than 15M states, for a total number of states estimated to be between 100M and 200M.

## 6 CONCLUSIONS AND OUTLOOK

We introduced a new notion of fixed point for zero-sum games that is restricted to a subset of the state-space, but still suffices to construct saddle-point and security policies. We then proposed the SPE algorithm that provably converges to a $Q$-function that satisfies the restricted fixed-point condition for deterministic finite games. The primary benefit of the restricted fixed-point condition is that convergence to a saddle-point can be achieved without full state exploration, which was required in previous works. Finally, we presented several numerical examples showing that, in practice, SPE consistently terminates at a saddle-point without exploring the entire state space. In fact, for several scalable board games, the fraction of states explored decreases as the game size increases. Importantly, we demonstrated that the $Q$-function used to construct the saddle-point can be represented either tabularly or using a neural network, without having to adapt SPE.

Our numerical results showed that some games permit termination at a saddle-point with a smaller fraction of explored states than others. Characterizing which classes of games are especially attractive from this perspective remains an important direction for future research. Non-cooperative games share similar structures with robust optimization, which should enable extending the ideas in this paper in that direction.

## 7    REPRODUCIBILITY STATEMENT

This work is reproducible primarily due to inclusion of all technical assumptions and proofs for the formal results in Sections 2-4, which are included either in the main text or in Appendices A.1-A.3. Towards the reproducibility of the numerical results, an anonymized version of the code is available at `https://anonymous.4open.science/r/saddlepointexploration-17BA/`. Importantly, all of the games that are used for illustrating the performance of the proposed methods are open-source (Lanctot et al., 2019; Darken, 2025).

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

## A  Technical Appendix

### A.1  Proof of Theorem 2

The following proposition is needed to prove Theorem 2. In stating this and subsequent results, we annotate equalities and inequalities involving random variables with $\overset{\text{wpo}}{=}$ or $\overset{\text{wpo}}{\geqslant}$ to indicate that they hold with probability one over the full randomness of the state-action-reward trajectory. This proposition is not really new, but we state it here (and provide a self-contained proof in Appendix A.2) because we could not find a version of it that matches the zero-sum turn games setup.

**Proposition 1.** *The following three statements hold for any pair of policies $(\pi_1, \pi_2) \in \Pi_1 \times \Pi_2$ and any absolutely summable function $V : \mathcal{S} \to \mathbb{R}$:*

1. *If*

$$V(s_t) \overset{\text{wpo}}{=} \mathrm{E}_{\pi_1,\pi_2}[r_{t+1} + \mathrm{sgn}_1(s_t)\,\mathrm{sgn}_1(s_{t+1})V(s_{t+1}) \mid s_t], \quad \forall t \geqslant 0, \qquad (24)$$

   *then*

$$V(s_\tau) \overset{\text{wpo}}{=} V_{\pi_1,\pi_2}(s_\tau), \quad \forall \tau \geqslant 0. \qquad (25)$$

2. *If*

$$\mathrm{sgn}_1(s_\tau)V(s_\tau) \overset{\text{wpo}}{\leqslant} \mathrm{E}_{\pi_1,\pi_2}\big[\,\mathrm{sgn}_1(s_\tau)r_{t+1} + \mathrm{sgn}_1(s_{t+1})V(s_{t+1}) \mid s_\tau\big], \quad \forall t \geqslant 0, \quad (26)$$

   *then*

$$\mathrm{sgn}_1(s_\tau)V(s_\tau) \overset{\text{wpo}}{\leqslant} \mathrm{sgn}_1(s_\tau)V_{\pi_1,\pi_2}(s_\tau), \quad \forall \tau \geqslant 0. \qquad (27)$$

3. *If*

$$\mathrm{sgn}_1(s_\tau)V(s_\tau) \overset{\text{wpo}}{\geqslant} \mathrm{E}_{\pi_1,\pi_2}\big[\,\mathrm{sgn}_1(s_\tau)r_{t+1} + \mathrm{sgn}_1(s_{t+1})V(s_{t+1}) \mid s_\tau\big], \quad \forall t \geqslant 0, \quad (28)$$

   *then*

$$\mathrm{sgn}_1(s_\tau)V(s_\tau) \overset{\text{wpo}}{\geqslant} \mathrm{sgn}_1(s_\tau)V_{\pi_1,\pi_2}(s_\tau), \quad \forall \tau \geqslant 0. \qquad (29)$$

$\square$

We are now ready to prove Theorem 2:

*Proof of Theorem 2.* Combining the definition of $V^\dagger$ with the restricted fixed-point condition, we obtain

$$V^\dagger(s) = \max_{a \in \mathcal{A}} Q^\dagger(s, a)$$

$$= \max_{a \in \mathcal{A}} \mathrm{E}\big[r_{t+1} + \mathrm{sgn}_1(s_t)\,\mathrm{sgn}_1(s_{t+1})V^\dagger(s_{t+1}) \mid s_t = s, a_t = a\big], \quad \forall s \in \mathcal{S}_{\Pi^\dagger}. \qquad (30)$$

When $s \in \mathcal{S}_1$, (14) guarantees that the policy $\pi_1^\dagger(s)$ reaches the maximum in (30), but an arbitrary policy $\pi_1(s)$ may not and therefore

$$V^\dagger(s) = \mathrm{E}[r_{t+1} + \mathrm{sgn}_1(s_t)\,\mathrm{sgn}_1(s_{t+1})V^\dagger(s_{t+1}) \mid s_t = s, a_t = \pi_1^\dagger(s)]$$

$$= \mathrm{E}_{\pi_1^\dagger \pi_2}[r_{t+1} + \mathrm{sgn}_1(s_t)\,\mathrm{sgn}_1(s_{t+1})V^\dagger(s_{t+1}) \mid s_t = s] \qquad (31)$$

$$\geqslant \mathrm{E}[r_{t+1} + \mathrm{sgn}_1(s_t)\,\mathrm{sgn}_1(s_{t+1})V^\dagger(s_{t+1}) \mid s_t = s, a_t = \pi_1(s)]$$

$$= \mathrm{E}_{\pi_1 \pi_2}[r_{t+1} + \mathrm{sgn}_1(s_t)\,\mathrm{sgn}_1(s_{t+1})V^\dagger(s_{t+1}) \mid s_t = s], \quad \forall s \in \mathcal{S}_1 \cap \mathcal{S}_{\Pi^\dagger}, \forall \pi_1, \pi_2, \quad (32)$$

where $\pi_2$ in (32) can be any policy in $\Pi_2$, because, for a state $s \in \mathcal{S}_1$, the policy of $\mathrm{P}_2$ makes no difference.

In contrast, when $s \in \mathcal{S}_2$, the policy $\pi_2^\dagger(s)$ reaches the maximum in (30), but an arbitrary policy $\pi_2(s)$ may not and we have

$$V^\dagger(s) = \mathrm{E}_{\pi_1 \pi_2^\dagger}[r_{t+1} + \mathrm{sgn}_1(s_t)\,\mathrm{sgn}_1(s_{t+1})V^\dagger(s_{t+1}) \mid s_t = s] \qquad (33)$$

$$\geqslant \mathrm{E}_{\pi_1 \pi_2}[r_{t+1} + \mathrm{sgn}_1(s_t)\,\mathrm{sgn}_1(s_{t+1})V^\dagger(s_{t+1}) \mid s_t = s], \quad \forall s \in \mathcal{S}_2 \cap \mathcal{S}_{\Pi^\dagger}, \forall \pi_1, \pi_2. \quad (34)$$

From (31) with $\pi_2 = \pi_2^\dagger$ and (33) with $\pi_1 = \pi_1^\dagger$, we obtain

$$V^\dagger(s) = \mathrm{E}_{\pi_1^\dagger \pi_2^\dagger}[r_{t+1} + \mathrm{sgn}_1(s_t)\,\mathrm{sgn}_1(s_{t+1})V^\dagger(s_{t+1}) \mid s_t = s], \quad \forall s \in \mathcal{S}_{\Pi^\dagger}.$$

Since the pair $(\pi_1^\dagger, \pi_2^\dagger)$ belongs to $\Pi^\dagger$, every trajectory generated under these policies belongs to $\mathcal{S}_{\Pi^\dagger}$ with probability 1, which enable us to use Proposition 1 to conclude that

$$V^\dagger(s) = V_{\pi_1^\dagger, \pi_2^\dagger}(s), \quad \forall s \in \mathcal{S}. \tag{35}$$

Multiplying (31) by $\mathrm{sgn}_1(s) = 1$, $\forall s \in \mathcal{S}_1 \cap \mathcal{S}_{\Pi^\dagger}$ and combining this equality with (34) multiplied by $\mathrm{sgn}_1(s) = -1$, $\forall s \in \mathcal{S}_2 \cap \mathcal{S}_{\Pi^\dagger}$ and with $\pi_1 = \pi_1^\dagger$, we obtain

$$\mathrm{sgn}_1(s)V^\dagger(s) \leqslant \mathrm{E}_{\pi_1^\dagger \pi_2}[r_{t+1}\,\mathrm{sgn}_1(s_t) + \mathrm{sgn}_1(s_{t+1})V^\dagger(s_{t+1}) \mid s_t = s], \quad \forall s \in \mathcal{S}_{\Pi^\dagger}.$$

Since every pair $(\pi_1^\dagger, \pi_2)$, $\forall \pi_2 \in \Pi_2$ also belongs to $\Pi^\dagger$ the trajectories generated under these policies belongs to $\mathcal{S}_{\Pi^\dagger}$ with probability 1, which enable us again to use Proposition 1 and now conclude that

$$\mathrm{sgn}_1(s)V^\dagger(s) \leqslant \mathrm{sgn}_1(s)V_{\pi_1^\dagger, \pi_2}(s), \quad \forall s \in \mathcal{S}_{\Pi^\dagger}.$$

Combining this inequality with (35) and using the fact that $\mathrm{sgn}_1(s) = -\mathrm{sgn}_2(s)$, $\forall s \in \mathcal{S}$, leads to

$$\mathrm{sgn}_1(s)V^\dagger(s) = \mathrm{sgn}_1(s)V_{\pi_1^\dagger, \pi_2^\dagger}(s) \leqslant \mathrm{sgn}_1(s)V_{\pi_1^\dagger, \pi_2}(s), \quad \forall s \in \mathcal{S}_{\Pi^\dagger}$$
$$\mathrm{sgn}_2(s)V^\dagger(s) = \mathrm{sgn}_2(s)V_{\pi_1^\dagger, \pi_2^\dagger}(s) \geqslant \mathrm{sgn}_2(s)V_{\pi_1^\dagger, \pi_2}(s), \quad \forall s \in \mathcal{S}_{\Pi^\dagger}. \tag{36}$$

If instead we multiply (33) by $\mathrm{sgn}_1(s) = -1$, $s \in \mathcal{S}_2 \cap \mathcal{S}_{\Pi^\dagger}$ and combine this equality with (32) multiplied by $\mathrm{sgn}_1(s) = 1$, $s \in \mathcal{S}_1 \cap \mathcal{S}_{\Pi^\dagger}$ and with $\pi_2 = \pi_2^\dagger$, we obtain

$$\mathrm{sgn}_1(s)V^\dagger(s) \geqslant \mathrm{E}_{\pi_1 \pi_2^\dagger}[r_{t+1}\,\mathrm{sgn}_1(s_t) + \mathrm{sgn}_1(s_{t+1})V^\dagger(s_{t+1}) \mid s_t = s], \quad \forall s \in \mathcal{S}_{\Pi^\dagger}.$$

Since every pair $(\pi_1, \pi_2^\dagger)$, $\forall \pi_2 \in \Pi_2$ also belongs to $\Pi^\dagger$ the trajectories generated under these policies belongs to $\mathcal{S}_{\Pi^\dagger}$ with probability 1 and we can again use Proposition 1 and (35) to conclude that

$$\mathrm{sgn}_1(s)V^\dagger(s) = \mathrm{sgn}_1(s)V_{\pi_1^\dagger, \pi_2^\dagger}(s) \geqslant \mathrm{sgn}_1(s)V_{\pi_1, \pi_2^\dagger}(s), \quad \forall s \in \mathcal{S}_{\Pi^\dagger}. \tag{37}$$

The saddle-point inequalities (6) follow from (36) and (37). ∎

## A.2 Proof of Proposition 1

*Proof of Proposition 1.* To prove the first statement, we multiply both sides of (24) by $\mathrm{sgn}_1(s_t)$, to conclude that

$$\mathrm{sgn}_1(s_t)V(s_t) - \mathrm{E}[\mathrm{sgn}_1(s_{t+1})V(s_{t+1}) \mid s_t] \stackrel{\mathrm{wpo}}{=} \mathrm{E}[r_{t+1}\,\mathrm{sgn}_1(s_t) \mid s_t], \quad \forall t \geqslant 0. \tag{38}$$

Suppose now that we pick some $\tau \geqslant 0$ and take conditional expectations of both sides of (38) given $s_\tau$. By the smoothing property of conditional expectations, we conclude that

$$\mathrm{E}[\mathrm{sgn}_1(s_t)V(s_t) \mid s_\tau] - \mathrm{E}[\mathrm{sgn}_1(s_{t+1})V(s_{t+1}) \mid s_\tau] \stackrel{\mathrm{wpo}}{=} \mathrm{E}[r_{t+1}\,\mathrm{sgn}_1(s_t) \mid s_\tau].$$

Adding both sides of this equality from $t = \tau$ to $t \to \infty$ and using the absolute convergence of the two series on the left-hand side, obtain

$$\mathrm{sgn}_1(s_\tau)V(s_\tau) \stackrel{\mathrm{wpo}}{=} \sum_{t=\tau}^{\infty} \mathrm{E}[r_{t+1}\,\mathrm{sgn}_1(s_t) \mid s_\tau], \tag{39}$$

from which (25) follows by multiplying both sides of the equality above by $\mathrm{sgn}_1(s_\tau)$ and using the definition of the value function in (4).

The subsequent statements can be similarly derived, by starting with the inequalities (26) and (28), instead of the equality in (38). ∎

A.3 PROOF OF THEOREM 3

The statement that we get a pair of saddle-point policies is an almost direct consequence of the termination condition in line 22, because the code in lines 7–12 and 15–20 essentially solves the optimizations that appear in the definition of saddle-point in (6). The proof of termination in finite time is more involved and requires both Theorem 2 and the following result:

**Lemma 1** (Finite convergence over a subset of $\mathcal{S} \times \mathcal{A}$). *Assume that the state and action spaces are finite, the game is deterministic, terminates in finite time, and $\alpha_k = 1$, $\forall k \geqslant 0$. Let $\mathcal{Z}_\infty \subset \mathcal{S} \times \mathcal{A}$ denote the set of states-action pairs $(s_t, a_t)$ that appears infinitely many times in the samples in lines 24–26. Then the sequence $Q^k$ converges in a finite number of iterations to a function $Q^\dagger : \mathcal{S} \times \mathcal{A} \to \mathbb{R}$ for which (8) holds $\forall (s, a) \in \mathcal{Z}_\infty$.* $\qquad\square$

The following definitions and basic result will be used to prove Lemma 1: we say that a state $s \in \mathcal{S}$ is *recurrent* if there exists a finite sequence of actions that takes the state $s$ to itself with positive probability. States that are not recurrent are called transients and we denote the sets of transient and recurrent states by $\mathcal{S}_{\mathrm{recurrent}}$ and $\mathcal{S}_{\mathrm{transient}}$, respectively. For games with finite termination time, recurrent states must have zero reward and, in fact, must always be followed by states also with zero reward, as noted in the following proposition:

**Proposition 2.** *Consider a stationary Markov game with finite termination time and an arbitrary sequence of actions $a_0, a_1, \dots$. If the sequence of states $s_0, s_1, \dots$ and rewards $r_1, r_2, \dots$ can occur with positive probability, i.e.,*

$$p(s_{t+1}, r_{t+1} \mid s_t, a_t) > 0, \quad \forall t \geqslant 0, \tag{40}$$

*and if $s_\tau \in \mathcal{S}_{\mathrm{recurrent}}$ for some $\tau \geqslant 0$, then $r_{t+1} = 0$, $\forall t \geqslant \tau$.* $\qquad\square$

*Proof of Proposition 2.* Assume by contradiction that there exist times $t \geqslant \tau \geqslant 0$ for which $s_\tau \in \mathcal{S}_{\mathrm{recurrent}}$ and $r_{t+1} > 0$, which means that the sequence of actions $a_\tau, \dots, a_t$ takes the state from $s_\tau$ to $s_{t+1}$ and leads to the reward $r_{t+1} > 0$ with some positive probability, in the sense of (40).

Since $s_\tau \in \mathcal{S}_{\mathrm{recurrent}}$ there must also exist a (possibly quite different) finite sequence of actions $\bar{a}_\tau, \dots, \bar{a}_{\bar{t}}$ that takes the state back to $s_\tau$ at some time $\bar{t} > \tau$. Specifically, there exist associated sequences of states $\bar{s}_\tau, \dots, \bar{s}_{\bar{t}+1}$ and rewards $\bar{r}_\tau, \dots, \bar{r}_{\bar{t}+1}$ that satisfy

$$\bar{s}_\tau = \bar{s}_{\bar{t}+1} = s_\tau, \qquad p(\bar{s}_{t+1}, \bar{r}_{t+1} \mid \bar{s}_t, \bar{a}_t) > 0, \quad \forall t \in \{\tau, \dots, \bar{t}\}. \tag{41}$$

Since it is possible to return to the recurrent state $s_\tau$ as many times as we want, we can assume without loss of generality that $\bar{t}$ is larger than the termination time $T$, after which all rewards must be zero with probability one.

To complete the contradiction argument, we "concatenate" the above sequences of actions and states in the following order:

$$a_0, \dots, a_{\tau-1}, \qquad \bar{a}_\tau, \dots, \bar{a}_{\bar{t}}, \qquad a_\tau, \dots, a_t$$

$$\underbrace{s_0, \dots, s_{\tau-1},}_{\text{from original seq.}} \qquad \underbrace{s_\tau = \bar{s}_\tau, \dots, \bar{s}_{\bar{t}},}_{\text{from recurrence of } s_\tau} \qquad \underbrace{\bar{s}_{\bar{t}+1} = s_\tau, s_{\tau+1}, \dots, s_t}_{\text{back to original seq.}} .$$

All transitions in this sequence have positive probability because of either (40) or (41). In addition, the last state $s_t$ and action $a_t$ lead to a reward $r_{t+1} > 0$, which contradicts the fact that $\bar{t} > T$ and all rewards after that time must be zero with probability one. $\qquad\blacksquare$

*Proof of Lemma 1.* We now construct the function $Q^\dagger : \mathcal{S} \times \mathcal{A} \to \mathbb{R}$ to which $Q^k$ will converge. We start by setting

$$Q^\dagger(s, a) = 0, \quad \forall s \in \mathcal{S}_{\mathrm{recurrent}}, \ a \in \mathcal{A}.$$

In view of Proposition 2, at every recurrent state $s_t \in \mathcal{S}_{\mathrm{recurrent}}$ the reward $r_{t+1}$ must be equal to zero with probability one, regardless of the action $a_t$, and the same will happen for every subsequent reward $r_{\tau+1}$, $\forall \tau \geqslant t$. This means that, when we apply the update rule (11) for any state $s \in \mathcal{S}_{\mathrm{recurrent}}$

or for any state $s \in \mathcal{S}$ that can succeed a state in $\mathcal{S}_{\text{recurrent}}$ with positive probability, we will continue to have

$$Q^k(s, a) = 0, \quad \forall a \in \mathcal{A}, \ k \geqslant 0, . \tag{42}$$

This guarantees that $Q^\dagger$ will always match $Q^k$, at least over the set $\mathcal{S}_{\text{recurrent}} \times \mathcal{A}$.

Since $\mathcal{S} \times \mathcal{A}$ is finite and $\mathcal{Z}_\infty \subset \mathcal{S} \times \mathcal{A}$ includes all state-action pairs that appear infinitely many times in the samples in lines 24–26, there is going to exist a finite integer $K_0$ such that for every $k \geqslant K_0$ only states $(s_t, a_t) \in \mathcal{Z}_\infty$ appear in these samples. This means that we can define

$$Q^\dagger(s, a) = Q^{K_0}(s, a), \quad \forall (s, a) \notin \mathcal{Z}_\infty, \tag{43}$$

because after time $K_0$ no update of $Q^\dagger(s, a)$ outside $\mathcal{Z}_\infty$ will ever take place.

To complete the definition of $Q^\dagger(s, a)$ it now remains to define this function for pairs $(s, a) \in \mathcal{Z}_\infty$ with $s \in \mathcal{S}_{\text{transient}}$. To this effect, consider a directed graph $\mathcal{G}$ whose nodes are the transient states in $\mathcal{S}_{\text{transient}}$, with an edge from $s \in \mathcal{S}_{\text{transient}}$ to $s' \in \mathcal{S}_{\text{transient}}$ if there is an action $a \in \mathcal{A}$ for which a transition from $s$ to $s'$ is possible, i.e.,

$$\exists a \in \mathcal{A}, r \in \mathcal{R} : p(s', r | s, a) = 1.$$

This graph cannot have cycles because any nodes in a cycle would be recurrent and thus not in $\mathcal{S}_{\text{transient}}$. The absence of cycle guarantees that this graph has at least one topological ordering $\prec$, i.e., there exists a total order on the set $\mathcal{S}_{\text{transient}}$ transient states so that $s \prec s'$ if and only if the state $s$ cannot be reached from $s'$ (Cormen, 2009).

Let $s_{\max}$ be the "largest" state in $\mathcal{S}_{\text{transient}}$ with respect to the order $\prec$, i.e., $s_{\max} > s, \forall s \in \mathcal{S}_{\text{transient}} \backslash \{s_{\max}\}$. If for a given action $a \in \mathcal{A}$, we have that $(s_{\max}, a) \notin \mathcal{Z}_\infty$, convergence of $Q^k(s_{\max}, a)$ to (43) at iteration $K_0$ has already been established. If instead $(s_{\max}, a) \in \mathcal{Z}_\infty$, then $Q^k(s_{\max}, a)$ will eventually be updated using (11) at some finite iteration $k \geqslant K_0$. Moreover, since $s_{\max}$ is the "largest" state in $\mathcal{S}_{\text{transient}}$, it cannot transition to any transient state so it must necessarily transition to a recurrent state. In view of (42), the update in (11) with $\alpha_k = 1$ will necessarily be of the form

$$Q^{k+1}(s_{\max}, a) = r_{t+1} =: Q^\dagger(s_{\max}, a), \tag{44}$$

where $r_{t+1}$ is the (deterministic) reward arising from state $s_t = s_{\max}$ and action $a_t = a$. This means that every $Q^k(s_{\max}, a), a \in \mathcal{A}$ will converge to the value $Q^\dagger(s_{\max}, a)$ defined in (44), right after their first update.

We now use the order $>$ to build an induction argument showing that finite-time convergence will also happen for every other $s \in \mathcal{S}_{\text{transient}}$ and every $a \in \mathcal{A}$. To this effect, pick some $s \in \mathcal{S}_{\text{transient}}$, $a \in \mathcal{A}$ and assume by the induction hypothesis that every $Q^k(s', a)$ has converged to $Q^\dagger(s, a)$ for every $s' > s, a \in \mathcal{A}$ at some finite iteration $K_s \geqslant K_0$.

In case $(s, a) \notin \mathcal{Z}_\infty$, convergence of $Q^k(s_{\max}, a)$ to (43) at iteration $K_0$ has already been established. If instead $(s, a) \in \mathcal{Z}_\infty$, then $Q^k(s, a)$ will eventually be updated using (11) at some finite iteration $k \geqslant K_s$. At this iteration, the update in (11) with $\alpha_k = 1$ takes the form

$$Q^{k+1}(s, a) = r_{t+1} + \text{sgn}_1(s) \, \text{sgn}_1(s_{t+1}) \max_{a' \in \mathcal{A}} Q^k(s_{t+1}, a'),$$

where $r_{t+1}$ and $s_{t+1} > s$ are the (deterministic) reward and next state, respectively, arising from state $s_t = s$ and action $a_t = a$. But since $s_{t+1} > s$, the induction hypothesis guarantees that at this and at any subsequent update for the pair $(s, a)$, we have

$$Q^{k+1}(s, a) = r_{t+1} + \text{sgn}_1(s) \, \text{sgn}_1(s_{t+1}) \max_{a' \in \mathcal{A}} Q^\dagger(s_{t+1}, a') =: Q^\dagger(s, a). \tag{45}$$

This shows that every $Q^k(s, a), a \in \mathcal{A}$ will converge to the value $Q^\dagger(s, a)$ defined in (45), at their first update after $K_s$. By induction, and recursively defining $Q^\dagger(s, a)$ for every transient state $s$, we then conclude that $Q^k(s, a)$ converges to $Q^\dagger(s, a)$ also for every $s \in \mathcal{S}_{\text{transient}}, a \in \mathcal{A}$. ∎

The following result can be obtained by combining Lemma 1 and Theorem 2, when we freeze the policy of $\mathsf{P}_1$'s at $\pi_1^k$ and optimize over the policies of $\mathsf{P}_2$ or, alternatively, freeze the policy of $\mathsf{P}_2$'s at $\pi_2^k$ and optimize over the policies of $\mathsf{P}_1$. Note that when the policy of one of the players is "frozen" (i.e., not optimized) there is no distinction between restricted or regular fixed point. In this case, Lemma 1 guarantees convergence to a fixed point and Theorem 2 optimality (for the player that is not frozen).

**Corollary 1.** *Assume that the state and action spaces are finite and that the game is deterministic, terminates in finite time, the updates of $Q_2$ and $Q_1$ use $\alpha_k = 1$, and Assumption 1 holds. Then $\mathsf{P}_2$'s policy $\pi_2^{\mathrm{br}}$ (16) is optimal against $\mathsf{P}_1$'s policy $\pi_1^k$, with the rewards in (17), which means that*

$$\bar{J}_2(s_0) = \mathrm{sgn}_2(s_0) \max_{a \in \mathcal{A}} Q_2^{k_2}(s_0, a) = \mathrm{sgn}_2(s_0) V_{\pi_1^k, \pi_2^{\mathrm{br}}}(s_0) = \max_{\pi_2 \in \Pi_2} \mathrm{sgn}_2(s_0) V_{\pi_1^k, \pi_2}(s_0) \quad (46)$$

*and $\mathsf{P}_1$'s policy $\pi_1^{\mathrm{br}}$ (18) is optimal against $\mathsf{P}_2$'s policy $\pi_2^k$ with the rewards in (19), which means that*

$$\tilde{J}_1(s_0) = \mathrm{sgn}_1(s_0) \max_{a \in \mathcal{A}} Q_1^{k_1}(s_0, a) = \mathrm{sgn}_1(s_0) V_{\pi_1^{\mathrm{br}}, \pi_2^k}(s_0) = \max_{\pi_1 \in \Pi_1} \mathrm{sgn}_1(s_0) V_{\pi_1, \pi_2^k}(s_0). \quad (47)$$

$\square$

We are now ready to prove Theorem 3:

*Proof of Theorem 3.* Since $\mathrm{sgn}_2(s_0) = -\mathrm{sgn}_1(s_0)$, we conclude from (46) that

$$\mathrm{sgn}_2(s_0) \max_{a \in \mathcal{A}} Q_2^{k_2}(s_0, s) = \max_{\pi_2 \in \Pi_2} \mathrm{sgn}_2(s_0) V_{\pi_1^k, \pi_2}(s_0) \geqslant \max_{\pi_2 \in \Pi_2} \min_{\pi_1 \in \Pi_1} \mathrm{sgn}_2(s_0) V_{\pi_1, \pi_2}(s_0) \quad (48a)$$

$$\mathrm{sgn}_1(s_0) \max_{a \in \mathcal{A}} Q_2^{k_2}(s_0, a) = -\mathrm{sgn}_2(s_0) \max_{a \in \mathcal{A}} Q_2^{k_2}(s_0, a) = -\max_{\pi_2 \in \Pi_2} \mathrm{sgn}_2(s_0) V_{\pi_1^k, \pi_2}(s_0)$$

$$= \min_{\pi_2 \in \Pi_2} \mathrm{sgn}_1(s_0) V_{\pi_1^k, \pi_2}(s_0) \leqslant \max_{\pi_1 \in \Pi_1} \min_{\pi_2 \in \Pi_2} \mathrm{sgn}_1(s_0) V_{\pi_1, \pi_2}(s_0). \quad (48b)$$

Similarly, since $\mathrm{sgn}_2(s_0) = -\mathrm{sgn}_1(s_0)$, we conclude from (47) that

$$\mathrm{sgn}_1(s_0) \max_{a \in \mathcal{A}} Q_1^{k_1}(s_0, a) = \max_{\pi_1 \in \Pi_1} \mathrm{sgn}_1(s_0) V_{\pi_1, \pi_2^k}(s_0) \geqslant \max_{\pi_1 \in \Pi_1} \min_{\pi_2 \in \Pi_2} \mathrm{sgn}_1(s_0) V_{\pi_1, \pi_2}(s_0) \quad (49a)$$

$$\mathrm{sgn}_2(s_0) \max_{a \in \mathcal{A}} Q_1^{k_1}(s_0, a) = -\mathrm{sgn}_1(s_0) \max_{a \in \mathcal{A}} Q_1^{k_1}(s_0, a) = -\max_{\pi_1 \in \Pi_1} \mathrm{sgn}_1(s_0) V_{\pi_1, \pi_2^k}(s_0)$$

$$= \min_{\pi_1 \in \Pi_1} \mathrm{sgn}_2(s_0) V_{\pi_1, \pi_2^k}(s_0) \leqslant \max_{\pi_2 \in \Pi_2} \min_{\pi_1 \in \Pi_1} \mathrm{sgn}_2(s_0) V_{\pi_1, \pi_2}(s_0). \quad (49b)$$

The bounds in (20) follow from combining (48) with (49).

Upon termination, we have that

$$\mathrm{sgn}_2(s_0) \max_{a \in \mathcal{A}} Q_2^{k_2}(s_0, a) + \mathrm{sgn}_1(s_0) \max_{a \in \mathcal{A}} Q_1^{k_1}(s_0, a) \leqslant \eta$$

and, since $\mathrm{sgn}_2(s_0) = -\mathrm{sgn}_1(s_0)$, we conclude from (46) and (47) that

$$\max_{\pi_2 \in \Pi_2} \mathrm{sgn}_2(s_0) V_{\pi_1^k, \pi_2}(s_0) = \mathrm{sgn}_2(s_0) \max_{a \in \mathcal{A}} Q_2^{k_2}(s_0, a) \leqslant -\mathrm{sgn}_1(s_0) \max_{a \in \mathcal{A}} Q_1^{k_1}(s_0, a) + \eta$$

$$= -\max_{\pi_1 \in \Pi_1} \mathrm{sgn}_1(s_0) V_{\pi_1, \pi_2^k}(s_0) + \eta = \min_{\pi_1 \in \Pi_1} \mathrm{sgn}_2(s_0) V_{\pi_1, \pi_2^k}(s_0) + \eta.$$

Using the definition of $\max$ (on left-hand side) and of $\min$ (on right-hand side), we obtain

$$\mathrm{sgn}_2(s_0) V_{\pi_1^k, \pi_2^k}(s_0) \leqslant \max_{\pi_2 \in \Pi_2} \mathrm{sgn}_2(s_0) V_{\pi_1^k, \pi_2}(s_0)$$

$$= \mathrm{sgn}_2(s_0) \max_{a \in \mathcal{A}} Q_2^{k_2}(s_0, a) \leqslant -\mathrm{sgn}_1(s_0) \max_{a \in \mathcal{A}} Q_1^{k_1}(s_0, a) + \eta$$

$$= \min_{\pi_1 \in \Pi_1} \mathrm{sgn}_2(s_0) V_{\pi_1, \pi_2^k}(s_0) + \eta \leqslant \mathrm{sgn}_2(s_0) V_{\pi_1^k, \pi_2^k}(s_0) + \eta,$$

from which we conclude that

$$\mathrm{sgn}_2(s_0) V_{\pi_1^k, \pi_2^k}(s_0) \geqslant \max_{\pi_2 \in \Pi_2} \mathrm{sgn}_2(s_0) V_{\pi_1^k, \pi_2}(s_0) - \eta$$

and also that

$$-\mathrm{sgn}_2(s_0) V_{\pi_1^k, \pi_2^k}(s_0) \geqslant -\min_{\pi_1 \in \Pi_1} \mathrm{sgn}_2(s_0) V_{\pi_1, \pi_2^k}(s_0, a) - \eta$$

$$\Leftrightarrow \quad \mathrm{sgn}_1(s_0) V_{\pi_1^k, \pi_2^k}(s_0) \geqslant \max_{\pi_1 \in \Pi_1} \mathrm{sgn}_1(s_0) V_{\pi_1, \pi_2^k}(s_0, a) - \eta,$$

from which we conclude that $(\pi_1^k, \pi_2^k)$ is an $\eta$-saddle-point.

Having established that the algorithm can only terminate at an $\eta$-saddle point, it remains to prove it always terminates in finite time. By contradiction, assume that the algorithm does not terminate. Since the state and action sets are finite, the number of pairs of deterministic policies in $\Pi_1 \times \Pi_2$ policies is also finite, which means that there must then exist at least one pair of policies $(\pi_1^\dagger, \pi_2^\dagger)$ for which the pairs $(\pi_1^k, \pi_2^k)$ defined in lines 7 and 15 turns out to be equal to $(\pi_1^\dagger, \pi_2^\dagger)$ infinitely many times.

Suppose now that we define the following set of pairs of policies

$$\Pi^\dagger := \Pi_1^\dagger \cup \Pi_2^\dagger, \qquad \Pi_1^\dagger := \{(\pi_1^\dagger, \pi_2) : \pi_2 \in \Pi_2\}, \qquad \Pi_2^\dagger := \{(\pi_1, \pi_2^\dagger) : \pi_1 \in \Pi_1\}.$$

Verifying in line 12 that $Q_2^{k_2}(\cdot, \cdot)$ has converged, requires the set of samples in line 9 to include every pair $(s_t, a_t)$ in $\mathcal{S}_{\Pi_1^\dagger} \times \mathcal{A}$, whereas verifying in line 20 that $Q_1^{k_1}(\cdot, \cdot)$ has converged, requires the set of samples in line 17 to include every pair $(s_t, a_t)$ in $\mathcal{S}_{\Pi_2^\dagger} \times \mathcal{A}$. This means that each of the updates of $Q^k$ in lines 24–26 will include, at least, every pair $(s_t, a_t)$ in $\mathcal{S}_{\Pi^\dagger} \times \mathcal{A}$ infinitely many times. This enable us to then apply Lemma 1 to conclude that, $Q^k$ must converge in a finite number of iterations to a function $Q^\dagger : \mathcal{S} \times \mathcal{A} \to \mathbb{R}$ for which (8) holds $\forall (s, a) \in \mathcal{Z}_\infty \supset \mathcal{S}_{\Pi^\dagger} \times \mathcal{A}$.

We have just established that the function $Q^\dagger : \mathcal{S} \times \mathcal{A} \to \mathbb{R}$ satisfies the assumptions of Theorem 2, from which we conclude that $(\pi_1^\dagger, \pi_2^\dagger)$ must be a saddle point policy that was reached by $(\pi_1^k, \pi_2^k)$ at some finite iteration $k$. This establishes a contradiction, because as soon as $(\pi_1^k, \pi_2^k)$ reaches a saddle-point the algorithm will terminate. ∎

### A.4 Proof of Theorem 4

To extend Lemma 1 to the stochastic case, we start with a zero initial estimate $Q^0 : \mathcal{S} \times \mathcal{A} \to \mathbb{R}$, $Q^0(s, a) = 0$, $\forall s \in \mathcal{S}$, $\forall a \in \mathcal{A}$, and iteratively draws samples $\{(s_{t_k}, a_{t_k}, s_{t_k+1}, r_{t_k+1}) : \forall k \geqslant 0\}$ from the transition/reward probability function $p(s_{t_k+1}, r_{t_k+1} \mid s_{t_k}, a_{t_k})$, leading to an update of the form

$$Q^{k+1}(s, a) = \begin{cases} (1 - \alpha_k)Q^k(s, a) + \alpha_k Q_{\text{target}}^{k+1} & s = s_{t_k}, a = a_{t_k}, \\ Q^k(s, a) & \text{otherwise,} \end{cases} \qquad \forall s \in \mathcal{S}, \ a \in \mathcal{A}, \quad (50)$$

for some sequence $\alpha_k \in (0, 1)$ and

$$Q_{\text{target}}^{k+1} := r_{t_k+1} + \text{sgn}_1(s_{t_k}) \, \text{sgn}_1(s_{t_k+1}) \max_{a' \in \mathcal{A}} Q^k(s_{t_k+1}, a').$$

Defining the indicator function

$$I_{s,a}(\bar{s}, \bar{a}) = \begin{cases} 1 & \bar{s} = s, \ \bar{a} = a \\ 0 & \text{otherwise.} \end{cases}$$

and the (random) sequences

$$\{\bar{\alpha}_k(s, a) := \alpha_k I_{s,a}(s_{t_k}, a_{t_k}) : \forall k \geqslant 0\}, \qquad \forall s \in \mathcal{S}, \ a \in \mathcal{A},$$

we can compactly re-write (50) as

$$Q^{k+1}(s, a) = (1 - \bar{\alpha}_k(s, a))Q^k(s, a) + \bar{\alpha}_k(s, a)Q_{\text{target}}^{k+1}, \qquad \forall s \in \mathcal{S}, \ a \in \mathcal{A}.$$

**Lemma 2** (Finite convergence over a subset of $\mathcal{S} \times \mathcal{A}$: Stochastic Case). *Assume that the state and action spaces are finite and that the game terminates in finite time. Let $\mathcal{Z}_\infty \subset \mathcal{S} \times \mathcal{A}$ denote the (random) set of states-action pairs $(s_t, a_t)$ that appears infinitely many times in the samples in lines 24–26 and assume that the sequences $\{\bar{\alpha}_k(s, a) \in (0, 1) : \forall k \geqslant 0\}$, $s \in \mathcal{S}, a \in \mathcal{A}$ satisfy*

$$P\left(\sum_{t=0}^\infty \bar{\alpha}_k(s, a) = +\infty, \quad \sum_{t=0}^\infty \bar{\alpha}_k^2(s, a) < \infty, \quad \forall (s, a) \in \mathcal{Z}_\infty\right) = 1.$$

*Then the sequence $Q^k$ converges with probability one and*

$$P\left(\lim_{k \to \infty} Q^k = Q^\dagger, \text{ and } (8) \text{ holds for } Q^\dagger, \forall (s, a) \in \mathcal{Z}_\infty\right) = 1. \qquad \square$$

It is important to note that, while Lemma 2 guarantes convergence to a function $Q^\dagger$ for which (8) holds on $\mathcal{Z}_\infty$, different realizations of the random variables will typically lead to distinct sets $\mathcal{Z}_\infty$ and to different limits $Q^\dagger$ and that these $Q^\dagger$ will generally not satisfy (8) outside $\mathcal{Z}_\infty$.

The proof of Lemma 2 combines elements from the proof of Lemma 1 with a stochastic approximation argument, which needs the following result.

**Lemma 3** (Restricted contraction). *Assume that the state and action spaces are finite and that the game terminates in finite time. Given a set $\mathcal{Z}_\infty \subset \mathcal{S} \times \mathcal{A}$ and a function $Q^\perp : \mathcal{S} \times \mathcal{A} \to \mathbb{R}$ with the property that*

$$Q^\perp(s, a) = 0, \quad \forall s \in \mathcal{S}_{\text{recurrent}}, a \in \mathcal{A}, \tag{51}$$

*consider the operator $F$ that maps a function $Q : \mathcal{S} \times \mathcal{A} \to \mathbb{R}$ to another function $F[Q] : \mathcal{S} \times \mathcal{A} \to \mathbb{R}$ defined by*

$$F[Q](s, a) = \begin{cases} \mathrm{E}\left[r_{t+1} + \mathrm{sgn}_1(s_t)\,\mathrm{sgn}_1(s_{t+1}) \max_{a' \in \mathcal{A}} Q(s_{t+1}, a') \,\big|\, s_t = s, a_t = a\right], & (s, a) \in \mathcal{Z}_\infty, s \notin \mathcal{S}_{\text{recurrent}} \\ Q^\perp(s, a) & \text{otherwise.} \end{cases} \tag{52}$$

*The set of functions*

$$\mathcal{Q}^\perp \coloneqq \{Q : \mathcal{S} \times \mathcal{A} \to \mathbb{R} : Q(s, a) = Q^\perp(s, a), \forall (s, a) \notin \mathcal{Z}_\infty\}, \tag{53}$$

*that match the value of $Q^\perp$ on $\mathcal{Z}_\infty$ is invariant under $F$ and $F$ is a contraction on $\mathcal{Q}^\perp$, i.e.,*

$$F(\mathcal{Q}^\perp) \subset \mathcal{Q}^\perp, \qquad \|F(Q_1 - Q_2)\| \leqslant \gamma \|Q_1 - Q_2\|, \quad \forall Q_1, Q_2 \in \mathcal{Q}^\perp$$

*for a suitably selected constant $\gamma \in (0, 1)$ and norm $\|\cdot\|$.* $\qquad\square$

*Proof of Lemma 3.* $F$ is invariant on $\mathcal{Q}^\perp$ simply because $F[Q](s, a) = Q^\perp(s, a), \forall (s, a) \notin \mathcal{Z}_\infty$. In view of Proposition 2, the set of recurrent states $\mathcal{S}_{\text{recurrent}}$ is absorbing and has zero reward, regardless of the control action. This, together with the finite-time assumption, enable us to use (Tseng, 1990, Lemma 3) to conclude that there exists constants $\gamma \in [0, 1), \omega_s > 0, s \in \mathcal{S}$ such that

$$\sum_{s' \notin \mathcal{S}_{\text{recurrant}}} \mathrm{P}(s_{t+1} = s' | s_t = s, a_t = a) \omega_{s'} \leqslant \gamma \omega_s. \tag{54}$$

Pick two arbitrary functions $Q_1, Q_2 \in \mathcal{Q}^\perp$ and $(s, a) \in \mathcal{Z}_\infty, s \notin \mathcal{S}_{\text{recurrent}}$. From the definition of $F$, we conclude that

$$F[Q_1](s, a) - F[Q_2](s, a) =$$

$$= \mathrm{E}\left[\mathrm{sgn}_1(s_t)\,\mathrm{sgn}_1(s_{t+1})\left(\max_{a_1 \in \mathcal{A}} Q_1(s_{t+1}, a_1) - \max_{a_2 \in \mathcal{A}} Q_2(s_{t+1}, a_2)\right) \,\big|\, s_t = s, a_t = a\right]$$

$$= \sum_{s' \in \mathcal{S}} \mathrm{sgn}_1(s)\,\mathrm{sgn}_1(s')\left(\max_{a_1 \in \mathcal{A}} Q_1(s', a_1) - \max_{a_2 \in \mathcal{A}} Q_2(s', a_2)\right) \mathrm{P}(s_{t+1} = s' | s_t = s, a_t = a)$$

$$\leqslant \sum_{s' \in \mathcal{S}} \mathrm{sgn}_1(s)\,\mathrm{sgn}_1(s')\left(Q_1(s', a_{s'}) - Q_2(s', a_{s'})\right) \mathrm{P}(s_{t+1} = s' | s_t = s, a_t = a)$$

where

$$a_{s'} \in \begin{cases} \arg\max_{a_1 \in \mathcal{A}} Q_1(s', a_1)\Big) & \mathrm{sgn}_1(s)\,\mathrm{sgn}_1(s') = 1, \\ \arg\max_{a_2 \in \mathcal{A}} Q_2(s', a_2) & \mathrm{sgn}_1(s)\,\mathrm{sgn}_1(s') = -1. \end{cases}$$

Using (51) and (54), we further conclude that

$$F[Q_2](s, a) - F[Q_1](s, a)$$

$$\leqslant \sum_{s' \notin \mathcal{S}_{\text{recurrent}}} \mathrm{sgn}_1(s)\,\mathrm{sgn}_1(s')\left(Q_1(s', a_{s'}) - Q_2(s', a_{s'})\right) \mathrm{P}(s_{t+1} = s' | s_t = s, a_t = a)$$

$$= \sum_{s' \notin \mathcal{S}_{\text{recurrent}}} \mathrm{sgn}_1(s)\,\mathrm{sgn}_1(s') \frac{Q_1(s', a_{s'}) - Q_2(s', a_{s'})}{\omega_{s'}} \mathrm{P}(s_{t+1} = s' | s_t = s, a_t = a) \omega_{s'}$$

$$\leqslant \gamma \omega_s \max_{s' \notin \mathcal{S}_{\text{recurrent}}} \frac{1}{\omega_{s'}} |Q_1(s', a_{s'}) - Q_2(s', a_{s'})|.$$

Similarly, we can also conclude that

$$F[Q_1](s,a) - F[Q_2](s,a) \leqslant \gamma\omega_s \max_{s'\notin\mathcal{S}_{\text{recurrent}}} \frac{1}{\omega_{s'}}|Q_1(s',a_{s'}) - Q_2(s',a_{s'})|,$$

from which we obtain the bound

$$|F[Q_1](s,a) - F[Q_2](s,a)| \leqslant \gamma\omega_s \max_{s'\notin\mathcal{S}_{\text{recurrent}}} \frac{1}{\omega_{s'}}|Q_1(s',a_{s'}) - Q_2(s',a_{s'})|,$$

$\forall(s,a) \in \mathcal{Z}_\infty, s \notin \mathcal{S}_{\text{recurrent}}$. In view of the definition of $F$ and the set $\mathcal{Q}$, both the left and right-hand sides of the inequality above are equal to zero for all remaining pairs $s \in \mathcal{S}, a \in \mathcal{A}$, which means that this inequality actually holds over $\mathcal{S} \times \mathcal{A}$. Dividing both sides of the inequality by $\omega_s$ and taking a maximum over $\mathcal{S} \times \mathcal{A}$, we conclude that

$$\max_{s\in\mathcal{S}}\max_{a\in\mathcal{A}} \frac{1}{\omega_s}|F[Q_1](s,a) - F[Q_2](s,a)| \leqslant \gamma \max_{s'\in\mathcal{S}}\max_{a\in\mathcal{A}} \frac{1}{\omega_{s'}}|Q_1(s',a) - Q_2(s',a)|,$$

which confirms that $F$ is a contraction on $\mathcal{Q}^\perp$, with respect to an $L_\infty$-norm weighted by the scalars $\omega_s$. ∎

*Proof of Lemma 2.* Proceeding exactly as in the proof of Lemma 1, we conclude that

$$Q^k(s,a) = 0, \quad \forall(s,a) \in \mathcal{S}_{\text{recurrent}} \times \mathcal{A}, \ k \geqslant 0 \tag{55}$$

and also that there exists a finite integer $K_0$ such that $(s_{t_k}, a_{t_k}) \in \mathcal{Z}_\infty$, for every $k \geqslant K_0$. This means that we can define

$$Q^\dagger(s,a) = Q^{K_0}(s,a), \quad \forall(s,a) \notin \mathcal{Z}_\infty \text{ or } (s,a) \in \mathcal{S}_{\text{recurrent}} \times \mathcal{A} \tag{56}$$

because after time $K_0$ no update of $Q^\dagger(s,a)$ outside $\mathcal{Z}_\infty$ will ever take place. The remainder of this proof uses an stochastic approximation argument to show that the values of $Q^k(s,a)$ also converge for the pairs $(s,a) \in \mathcal{Z}_\infty, s \notin \mathcal{S}_{\text{recurrent}}$.

In general, the set $\mathcal{Z}_\infty$, the integer $K_0$, and the function $Q^{K_0}$ are all random. In the remainder of the proof, we fix a realization for $\mathcal{Z}_\infty, K_0, Q^{K_0}$ that can occur with positive probability and condition all probabilities to this realization. To emphasize this, we use a subscript $\mathrm{E}_{\mathcal{Z}_\infty,K_0,Q^{K_0}}$ in the expected value operator. For statements that occur "with probability one" this is not needed, since conditioning to a positive probability event will not affect whether or not the statement holds with probability one.

We proceed by re-writing (50) as

$$Q^{k+1}(s,a) = (1 - \bar{\alpha}_k)Q^k(s,a) + \bar{\alpha}_k(F_{\mathcal{Z}_\infty,K_0,Q^{K_0}}[Q^k](s_{t_k},a_{t_k}) + w_k)$$

where $F_{\mathcal{Z}_\infty,K_0,Q^{K_0}}$ is the operator defined in (52) and

$$w_k := Q_{\text{target}}^{k+1} - F_{\mathcal{Z}_\infty,K_0,Q^{K_0}}[Q^k](s_{t_k},a_{t_k}).$$

The subscript in $F_{\mathcal{Z}_\infty,K_0,Q^{K_0}}$ emphasizes that, for the purposes of this proof, the expectation in (52) should be understood as conditioned to a positive probability realization of $\mathcal{Z}_\infty, K_0, Q^{K_0}$.

For $k \geqslant K_0$ and $s_{t_k} \notin \mathcal{S}_{\text{recurrent}}$, the value of $F_{\mathcal{Z}_\infty,K_0,Q^{K_0}}[Q^k](s_{t_k},a_{t_k})$ is defined by the top branch in (52) and we have that

$$F_{\mathcal{Z}_\infty,K_0,Q^{K_0}}[Q^k](s_{t_k},a_{t_k}) \overset{\text{wpo}}{=} \mathrm{E}_{\mathcal{Z}_\infty,K_0,Q^{K_0}}[Q_{\text{target}}^{k+1} \mid s_{t_k},a_{t_k}]$$

$$\Rightarrow \quad \mathrm{E}_{\mathcal{Z}_\infty,K_0,Q^{K_0}}[w_k \mid s_{t_k},a_{t_k}] \overset{\text{wpo}}{=} 0.$$

Alternatively, when $k \geqslant K_0$ but $s_{t_k} \in \mathcal{S}_{\text{recurrent}}$, we can conclude form Proposition 2 that all subsequent states belong to $\mathcal{S}_{\text{recurrent}}$ and all subsequent rewards are zero with probability one. Because of this and (55), we conclude that, also in this case, we have

$$F_{\mathcal{Z}_\infty,K_0,Q^{K_0}}[Q^k](s_{t_k},a_{t_k}) \overset{\text{wpo}}{=} \mathrm{E}_{\mathcal{Z}_\infty,K_0,Q^{K_0}}[Q_{\text{target}}^{k+1} \mid s_{t_k},a_{t_k}] = 0$$

$$\Rightarrow \quad \mathrm{E}_{\mathcal{Z}_\infty,K_0,Q^{K_0}}[w_k \mid s_{t_k},a_{t_k}] \overset{\text{wpo}}{=} 0.$$

In view of Lemma 3, the operator $F_{\mathcal{Z}_\infty,K_0,Q^{K_0}}$ is a contraction on $\mathcal{Q}^\perp$ defined in (53), which enable us to use (Tsitsiklis, 1994, Theorem 3) to conclude that $Q^k$ converges to a fixed-point of $F$ within $\mathcal{Q}$ and conclude the proof. In applying Lemma 3, all probabilities need to be conditioned to a specific positive probability realization of $\mathcal{Z}_\infty, K_0, Q^{K_0}$, but that does not invalidate the result. It does however mean that, different realizations may lead to different contraction constants $\gamma$. ∎

We are now ready to prove Theorem 4:

*Proof of Theorem 4.* Since $\mathrm{sgn}_2(s_0) = -\mathrm{sgn}_1(s_0)$, we conclude from (21) that

$$\mathrm{sgn}_2(s_0) \max_{a \in \mathcal{A}} Q_2^{k_2}(s_0, a) \geqslant \max_{\pi_2 \in \Pi_2} \mathrm{sgn}_2(s_0) V_{\pi_1^k, \pi_2}(s_0) - \epsilon \geqslant \max_{\pi_2 \in \Pi_2} \min_{\pi_1 \in \Pi_1} \mathrm{sgn}_2(s_0) V_{\pi_1, \pi_2}(s_0) - \epsilon$$
(57a)

$$\mathrm{sgn}_1(s_0) \max_{a \in \mathcal{A}} Q_2^{k_2}(s_0, a) = -\mathrm{sgn}_2(s_0) \max_{a \in \mathcal{A}} Q_2^{k_2}(s_0, a) \leqslant -\max_{\pi_2 \in \Pi_2} \mathrm{sgn}_2(s_0) V_{\pi_1^k, \pi_2}(s_0) + \epsilon$$
$$= \min_{\pi_2 \in \Pi_2} \mathrm{sgn}_1(s_0) V_{\pi_1^k, \pi_2}(s_0) + \epsilon \leqslant \max_{\pi_1 \in \Pi_1} \min_{\pi_2 \in \Pi_2} \mathrm{sgn}_1(s_0) V_{\pi_1, \pi_2}(s_0) + \epsilon.$$
(57b)

Similarly, since $\mathrm{sgn}_2(s_0) = -\mathrm{sgn}_1(s_0)$, we conclude from (22) that

$$\mathrm{sgn}_1(s_0) \max_{a \in \mathcal{A}} Q_1^{k_1}(s_0, a) \geqslant \max_{\pi_1 \in \Pi_1} \mathrm{sgn}_1(s_0) V_{\pi_1, \pi_2^k}(s_0) - \epsilon \geqslant \max_{\pi_1 \in \Pi_1} \min_{\pi_2 \in \Pi_2} \mathrm{sgn}_1(s_0) V_{\pi_1, \pi_2}(s_0) - \epsilon$$
(58a)

$$\mathrm{sgn}_2(s_0) \max_{a \in \mathcal{A}} Q_1^{k_1}(s_0, a) = -\mathrm{sgn}_1(s_0) \max_{a \in \mathcal{A}} Q_1^{k_1}(s_0, a) \leqslant -\max_{\pi_1 \in \Pi_1} \mathrm{sgn}_1(s_0) V_{\pi_1, \pi_2^k}(s_0) + \epsilon$$
$$= \min_{\pi_1 \in \Pi_1} \mathrm{sgn}_2(s_0) V_{\pi_1, \pi_2^k}(s_0) + \epsilon \leqslant \max_{\pi_2 \in \Pi_2} \min_{\pi_1 \in \Pi_1} \mathrm{sgn}_2(s_0) V_{\pi_1, \pi_2}(s_0) + \epsilon.$$
(58b)

The bounds in (23) follow from combining (57) with (58).

Upon termination, we have that

$$\mathrm{sgn}_2(s_0) \max_{a \in \mathcal{A}} Q_2^{k_2}(s_0, a) + \mathrm{sgn}_1(s_0) \max_{a \in \mathcal{A}} Q_1^{k_1}(s_0, a) \leqslant \eta$$

and, since $\mathrm{sgn}_2(s_0) = -\mathrm{sgn}_1(s_0)$, we conclude from (21) and (22) that

$$\max_{\pi_2 \in \Pi_2} \mathrm{sgn}_2(s_0) V_{\pi_1^k, \pi_2}(s_0) - \epsilon \leqslant \mathrm{sgn}_2(s_0) \max_{a \in \mathcal{A}} Q_2^{k_2}(s_0, a) \leqslant -\mathrm{sgn}_1(s_0) \max_{a \in \mathcal{A}} Q_1^{k_1}(s_0, a) + \eta$$
$$\leqslant -\max_{\pi_1 \in \Pi_1} \mathrm{sgn}_1(s_0) V_{\pi_1, \pi_2^k}(s_0) + \epsilon + \eta = \min_{\pi_1 \in \Pi_1} \mathrm{sgn}_2(s_0) V_{\pi_1, \pi_2^k}(s_0) + \epsilon + \eta.$$

Using the definition of $\max$ (on left-hand side) and of $\min$ (on right-hand side), we obtain

$$\mathrm{sgn}_2(s_0) V_{\pi_1^k, \pi_2^k}(s_0) - \epsilon \leqslant \max_{\pi_2 \in \Pi_2} \mathrm{sgn}_2(s_0) V_{\pi_1^k, \pi_2}(s_0) - \epsilon$$
$$\leqslant \mathrm{sgn}_2(s_0) \max_{a \in \mathcal{A}} Q_2^{k_2}(s_0, a) \leqslant -\mathrm{sgn}_1(s_0) \max_{a \in \mathcal{A}} Q_1^{k_1}(s_0, a) + \eta$$
$$\leqslant \min_{\pi_1 \in \Pi_1} \mathrm{sgn}_2(s_0) V_{\pi_1, \pi_2^k}(s_0, a) + \epsilon + \eta \leqslant \mathrm{sgn}_2(s_0) V_{\pi_1^k, \pi_2^k}(s_0) + \epsilon + \eta,$$

from which we conclude that

$$\mathrm{sgn}_2(s_0) V_{\pi_1^k, \pi_2^k}(s_0) \geqslant \max_{\pi_2 \in \Pi_2} \mathrm{sgn}_2(s_0) V_{\pi_1^k, \pi_2}(s_0) - 2\epsilon - \eta$$

and also that

$$-\mathrm{sgn}_2(s_0) V_{\pi_1^k, \pi_2^k}(s_0) \geqslant -\min_{\pi_1 \in \Pi_1} \mathrm{sgn}_2(s_0) V_{\pi_1, \pi_2^k}(s_0, a) - 2\epsilon - \eta$$
$$\Leftrightarrow \quad \mathrm{sgn}_1(s_0) V_{\pi_1^k, \pi_2^k}(s_0) \geqslant \max_{\pi_1 \in \Pi_1} \mathrm{sgn}_1(s_0) V_{\pi_1, \pi_2^k}(s_0, a) - 2\epsilon - \eta,$$

from which we conclude that $(\pi_1^k, \pi_2^k)$ is a $(2\epsilon + \eta)$-saddle-point.

To prove termination in finite time, assume by contradiction that the algorithm does not terminate. Since the state and action sets are finite, the number of pairs of deterministic policies in $\Pi_1 \times \Pi_2$ policies is also finite, which means that there must then exist at least one pair of policies $(\pi_1^\dagger, \pi_2^\dagger)$ for which the pairs $(\pi_1^k, \pi_2^k)$ defined in lines 7 and 15 turns out to be equal to $(\pi_1^\dagger, \pi_2^\dagger)$ infinitely many times. Assumption 3, then enable us to use Lemma 2, with the guarantee that $\mathcal{Z}_\infty \overset{\mathrm{wpo}}{\supset} \mathcal{S}_{\Pi^\dagger} \times \mathcal{A}$, to conclude that the sequence $Q^k$ converges to a function $Q^\dagger$, for which (8) holds $\forall (s, a) \in \mathcal{S}_{\Pi^\dagger} \times \mathcal{A}$ and

therefore $Q^\dagger$ is a restricted fixed point. In view of Theorem 2, this means that any policies $(\pi_1^\dagger, \pi_2^\dagger)$ that satisfy (14) are a saddle-point. While $Q^k$ may never reach $Q^\dagger$ for a finite iteration $k$, the policies $(\pi_1^k, \pi_2^k)$ derived from $Q^k$ using (15) will become saddle-point for after some sufficiently large (but finite) iteration $K$. This is because, the set of policies is finite.

Since for every $k \geqslant K$, every pair $(\pi_1^k, \pi_2^k)$ is a saddle-point, we conclude from the definition of saddle point and (6) that

$$\max_{\pi_1 \in \Pi_1} \mathrm{sgn}_1(s_0) V_{\pi_1, \pi_2^k}(s_0) + \max_{\pi_2 \in \Pi_2} \mathrm{sgn}_2(s_0) V_{\pi_1^k, \pi_2}(s_0) = 0$$

and, in view of Assumption 2 that

$$\left| \mathrm{sgn}_2(s_0) \max_{a \in \mathcal{A}} Q_2^{k_2}(s_0) + \mathrm{sgn}_1(s_0) \max_{a \in \mathcal{A}} Q_1^{k_1}(s_0) \right| \leqslant 2\epsilon$$

with probability larger than $1 - \delta > 0$. When $\eta > 2\epsilon$, this means that for each $k > K$ the algorithm will terminate with positive probability, from which we conclude that it will terminate in finite time with probability one; thus completing the contradiction argument.

To complete the proof, we need to compute the probability that the algorithms terminates at some finite iteration $k$ without satisfying the saddle-point condition. To this effect let $e_\ell, \ell \geqslant 1$ be a boolean random variable that is equal to 1 if the test in line 22 is executed at least $\ell$ times and the $\ell$th test results in termination, but the exit policies $(\pi_1^k, \pi_2^k)$ are not a saddle point (false positive). In terms of these variables, the probability that the algorithms terminates without satisfying the saddle-point condition is given by

$$\mathrm{P}(\exists i : e_i = 1) = \sup_{\ell \geqslant 0} y_\ell = \lim_{\ell \to \infty} y_\ell,$$

where

$$y_\ell := \mathrm{P}(\exists i \leqslant \ell : e_i = 1).$$

These probabilities can be computed recursively, defining $y_0 = 0$ and noting that for $\ell \geqslant 1$, we have that

$$\begin{aligned} y_\ell &= \mathrm{P}(e_\ell = 1, \ \forall i < \ell, e_i = 0) + \mathrm{P}(\exists i < \ell : e_i = 1) \\ &= \mathrm{P}(e_\ell = 1 | \forall i < \ell, e_i = 0) \, \mathrm{P}(\forall i < \ell, e_i = 0) + \mathrm{P}(\exists i < \ell : e_i = 1) \\ &= \mathrm{P}(e_\ell = 1 | \forall i < \ell, e_i = 0)(1 - y_{\ell-1}) + y_{\ell-1}. \end{aligned}$$

To get $e_\ell = 1$, the policies $(\pi_1^k, \pi_2^k)$ at the $\ell$th test must not be a saddle-point and yet trigger the exit condition. In view of Assumption 2, this can only happen with probability smaller than $\delta_\ell$ and therefore

$$y_\ell \leqslant \delta_\ell(1 - y_{\ell-1}) + y_{\ell-1} \leqslant \delta_\ell + y_{\ell-1}$$

Adding both sides from $\ell = 1$ to $\ell \to \infty$, we conclude that

$$\lim_{\ell \to \infty} y_\ell \leqslant \sum_{\ell=1}^{\infty} \delta_\ell,$$

which conclude the proof. ∎

## A.5 SPEEDING UP CONVERGENCE

As noted before, the samples $(s_t, a_t, s_{t+1}, r_{t+1})$ generated in lines 9 and 17 to verify the saddle-point condition suffice to guarantee convergence to a restricted fixed point. However, executing the code in lines 7–12 and 15–20 until convergence of $Q_1^k$ and $Q_2^k$ can be costly for games with large state-spaces. This motivates using an additional mechanism to approximately solve the optimizations in (16), (18); which does not need to be "sufficiently accurate" to certify that $(\pi_1^k, \pi_2^k)$ is a saddle-point but it is computationally much cheaper. Such a mechanism can be used in line 4 without compromising the guarantees provided by Theorem 3.

Procedure 2 provides such a mechanism by essentially replicating in line 4 what is done in lines 7–12 and 15–20 with additional tables $\hat{Q}_1, \hat{Q}_2$ that function within the scope of line 4:

1. Collect samples using $\hat{Q}_1$ and $\hat{Q}_2$ as in lines 7–12 and 15–20, but repeat the loops only over a finite number of iterations $L$, rather than waiting until convergence.

2. Only initialize $\hat{Q}_1$ and $\hat{Q}_2$ at the start of Algorithm 1 instead of re-initializing them before executing each check as in lines 7–12 and 15–20.

---

**Procedure 2** Optional procedure for line 4 of Algorithm 1, assuming an initialization $\hat{Q}_i^0(s, a) = 0$, $\forall s \in \mathcal{S}, a \in \mathcal{A}, i \in \{1, 2\}$ at the start of Algorithm 1.

---

1: extract $\mathsf{P}_1$'s policy $\pi_1^k$ from $Q^k$ using (15)
2: extract $\mathsf{P}_2$'s policy $\pi_2^k$ from $Q^k$ using (15)
3: **for** $L$ iterations **do**
4:     generate sample(s) $(s_t, a_t, s_{t+1}, r_{t+1})$ from (3), restricting $a_t = \pi_1^k(s_t)$ when $s_t \in \mathcal{S}_1$
5:     use sample(s) to update $\hat{Q}_2^k$ using (11)

6:     generate sample(s) $(s_t, a_t, s_{t+1}, r_{t+1})$ from (3), restricting $a_t = \pi_2^k(s_t)$ when $s_t \in \mathcal{S}_2$
7:     use sample(s) to update $\hat{Q}_1^k$ using (11)

---

