# OpenReview forum: "Reinforcement learning for saddle-point equilibria without full state exploration"
_ICLR.cc/2026/Conference — Submitted to ICLR 2026_

### Official Review · Reviewer_PmtW · 2025-10-20

**Soundness:** 3
**Presentation:** 3
**Contribution:** 3
**Rating:** 4
**Confidence:** 3

**Summary:**

This paper studies finding saddle-point equilibria in a zero-sum Markov Game. The paper shows that exact optimality can be achieved without global convergence of the Q-function—it suffices to converge locally on the reachable subspace where the policies interact strategically. Based on this new finding, the authors propose SPE. For deterministic finite games, the algorithm provably converges to a restricted fixed point and terminates in finite time (Theorem 3).

**Strengths:**

1. The authors propose a new fixed-point condition for the state–action value function. Unlike the standard Bellman equation (which must hold for all states), this condition only needs to hold on a restricted subset of the state space — specifically, the states reachable under the current candidate policies.

2. The authors propose the SPE algorithm and prove the convergence to a restricted fixed point and termination in finite time for deterministic finite games.

3. In all experiments, SPE reached a certified saddle-point without exploring the entire state space. As game size increased, the fraction of explored states decreased, showing strong scalability.

**Weaknesses:**

1. The main convergence theorem (Theorem 3) only holds for deterministic zero-sum turn-based games with finite termination. However, in many real-world applications, the game is stochastic.

2. Although the authors claim their algorithm does not need to search the whole state space, the paper does not derive theoretical sample complexity or convergence rates, showing their method is better than previous methods.

3. In the experiments, all test domains are turn-based, finite board games. Moreover, the experiments benchmark against a hypothetical lower bound rather than comparing to concrete algorithms

**Questions:**

Please see the weakness part for my concern.

---

> ### Author Response · Authors · 2025-11-21
> **Response to specific issues raised by reviewer PmtW**
>
> **Extension to stochastic games:** Please see general comments.
>
> **Extension beyond finite-time termination:** Please see response to reviewer M6nd.
>
> **Theoretical upper bound on states visited:** Please see response to reviewer M6nd.
>
> **Comparison with hypothetical lower bound:** We actually did compare our algorithms with respect to concrete algorithms and, in fact, that was how we determined the total number of states for the games. We initially tried using epsilon-greedy exploration, but it proved very hard to explore the full state-space so that we could "certify" the saddle-point; so we ended up using an exhaustive exploration.
>
> The key reason why we did not include comparisons with specific alternative algorithms is that most of the algorithms that come with theoretical guarantees rely on running sufficiently many iterations until a fixed-point for the Q-function is reached, which requires the full reachable state-space to be visited at least once, so plots for competing algorithms would all appear on the 1.0 horizontal line in the left and right plots in Figure 1.
>
> A notable exception might be the "Golf with Exploiter” algorithm. The sample complexity analysis of this algorithm is based on the "Bellman Eluder dimension of the game" and could conceivably not need full state/action exploration for special classes of games. However, this algorithm has several computational difficulties (acknowledged by the authors and noted in our related work section).

---

### Official Review · Reviewer_M6nd · 2025-10-30

**Soundness:** 2
**Presentation:** 2
**Contribution:** 2
**Rating:** 2
**Confidence:** 4

**Summary:**

This paper studies Q-learning in turn-based two-player Markov zero-sum games. The authors introduce the notion of restricted fixed points for an operator on the Q-function and show that a saddle point can be derived from such a fixed point. Furthermore, an advantage of the proposed algorithm is that it only explores a subset of the entire state space, leading to improved efficiency compared to previous Q-learning algorithms.

**Strengths:**

- The paper characterizes sufficient conditions under which the Q-learning operator is contractive, thereby establishing sufficient conditions to compute such restricted fixed point.
- The computation of those restricted fixed points only requires visiting a subset of the entire state space.
- Numerical experiments are provided to validate the theoretical results.

**Weaknesses:**

- The paper focuses exclusively on deterministic zero-sum Markov games, where given the state and action, the next state is deterministic. This setting is significantly simpler than standard stochastic games, in which one of the main challenges arises from the uncertainty in state transitions.

- Theorem 3 only holds for games that terminate in finite time (which is different from Markov games with finite horizon). This assumption appears quite restrictive as it imposes strong structure constraints on the reward function of each state. In standard RL literature, the $\gamma$ discount factor is commonly used to ensure the value function remains bounded (and thus is absolutely summable). it would be helpful for the authors to justify why they adopt this assumption instead of following the conventional discounted setting.

- Despite claiming that the explored state space is a subset of the entire state space, the paper provides no theoretical results on the upper bound of the number of states visited by the algorithm. Thus it is difficult to theoretically quantify the improved state dependency of the proposed algorithm.

- Only an asymptotic convergence guarantee is provided for the proposed algorithm.

**Questions:**

- In line 244-246, the authors state that every fixed point is a restricted fixed point but the converse is not true. Can the author provides a specific game example where the converse fails to hold?

- In line 296, the authors mention that SPE is applicable to general zero-sum turn games. Can the author specify with more details? Is there any convergence guarantee when the operator fails to be contractive?

---

> ### Author Response · Authors · 2025-11-21
> **Response to specific issues raised by reviewer M6nd**
>
> **Extension to stochastic games:** Please see general comments
>
> **Finite time and discounted cost:** There is no fundamental difference between our definition of finite time (a time after which the reward becomes zero with probability 1) and Markov games with a finite horizon (where the summation in (3) would end at some time T, rather than +\infty). To go from the finite horizon definition to ours, one would simply add "time" to the state, and force a transition to a "game-over" state with zero rewards when the horizon T is reached. We chose this particular definition because it avoids the need to explicitly parameterize the value function by time and thus leads to cleaner formulas, from our perspective.
>
> We focus on finite-time termination because it captures the “board-game-like” settings in which we are interested more faithfully than infinite-horizon discounted returns. However, our results would not change much if we were to introduce a discount factor \gamma^t in eq. 2:
> 1) we would need the usual \gamma factor multiplying the left-hand side Q in eq. 8, and,
> 2) as the reviewer noted, the "summability" assumption in Theorem 2, could be easier to verify (in fact not needed at all, if the per-state rewards are bounded).
>
> When the per-state rewards are bounded there is an even simpler approach: For every \chi>0, a game with infinite horizon but discounted costs, can always be "truncated" to a game with a sufficiently large but finite termination time T, so that the costs of the truncated and the original games differ by less than \chi, regardless of the policies used by the two players. In this case, using the results in our paper to compute an \epsilon-saddle-point for the truncated game, automatically gives us a (\epsilon+2\chi)-saddle-point to the original infinite-horizon discounted game. In view of this, there is relatively little loss of generality between our setup and games with discounted cost that may not terminate in finite time (at least for bounded per-state rewards).
>
> A summary of the observations made above has been added to section 2 of the revised manuscript.
>
> **Theoretical upper bound on states visited:** Indeed, the paper does not provide any upper bound on the number of states visited, because we believe that there is no non-trivial upper bound for general games.
>
> **Asymptotic convergence and Sample Complexity of Algorithm 1:** In view of the answer to the question above, a worst-case sample complexity bound will likely not improve upon existing bounds. However, it is still very important to provide an asymptotic convergence result because our SPE exploration algorithm violates a very basic assumption of previous exploration algorithms. Specifically, it does not guarantee that the sequence of samples will eventually cover the whole reachable subspace. Moreover, when we included in the samples collected in line 4 some form of epsilon-greedy policy (which would guarantee full coverage), we observed that the algorithm ends up "wasting samples" by exploring regions of the state-space that will eventually not be needed to certify the policy. Specifically, the plots in Figure 1 invariably look worse (i.e., larger fractions of the state space explored before a saddle-point can be found).
>
> **Question 1 (example):** Yes we can: an example of a restricted fixed point that is not a fixed point can be (rather trivially) obtained by picking any fixed point Q and replacing the values of Q outside the set \\(\mathcal{S}\_{\Pi^\*}\\) obtained for that Q by an arbitrary number, say 0. Such examples occurred essentially in every single numerical example presented in Section 5. This is because we can see in that section that large portions of the reachable space were never visited and therefore we ended up with functions Q that were zero (our initialization) at those states. However, for essentially all those states the actual fixed point should not have a zero value: it should have the value of the cost-to-go from that state. The reason why we still got a saddle-point policy is that (i) either the value was "good enough" to generate a policy that met the exit condition or (ii) the state did not belong to \\(\mathcal{S}\_{\Pi^\*}\\). For all states that were never visited, the reason had to be (ii).
>
> **Question 2 (applicability of Algorithm 1 to general games):** The phrase was ambiguous (and has been fixed): we meant to say that the algorithm is applicable to stochastic games (not necessarily with finite termination time nor finite state/action spaces), but the remainder of the section was focused on deterministic games with finite state-space.
>
> Regarding the possibility of extending the analysis when the operator is not contractive, at this time we do not see a way around it. To be specific, our contribution is not on how to deal with cases where the operator is not a contraction, it is about certifying policies as saddle-point without converging to the fixed point of the operator.

---

> > ### Comment · Reviewer_M6nd · 2025-11-26
> >
> > I thank the authors for their clarifications. I still have several additional questions.
> >
> > - Regarding the way of transferring Markov games with finite horizon to finite time termination, once we add time $t$ to every state, would this expands the overall state space?
> >
> > - Furthermore, the recurrent states of the original game would be transient after the transition. Can the author elaborate more on the difference between recurrent states defined in Proposition 2 and the recurrent states in standard Markov games?
> >
> > - For question one, I do not get "replacing the values of $Q$ outside the set $S_{\Pi^*}$ by any arbitrary number". Could the author provide an explicit small example to show this?

---

> > > ### Author Response · Authors · 2025-11-28
> > >
> > > **Adding time to the state:** Adding time often increases the size of the state space, but not always. E.g., in Tic-Tac-Toe the number of X's plus the number of O's already "encodes time" and in Dots-And-Boxes the total number of lines in the board also "encodes time". However, in Atlatl the board configuration does not encode time because the blue player could move a piece forward, red could pass, and then  blue could move the same piece back to the start position: in this case we need to explicitly add time to the state to figure out that we are at time 2, rather than at time 0.
> > >
> > > However, it should not be too surprising if we needed to increase the number of states to solve a finite horizon game, because in such games the value function is not stationary and we need to compute
> > > 	$$V\_t(s) = \text{cost-to-go from }s\_t=s, \quad \forall t,s$$ (A)
> > > in our formulation we would encode time as (t,s) and instead define
> > > 	$$V(t,s) = \text{cost-to-go from }s\_t=s, \quad \forall t,s$$ (B)
> > > The option (B) has the advantage that it saves memory for games like Tic-Tac-Toe and Dots-And-Boxes since the reachable state-space of $(t,s)$ has the same number of elements as the reachable state-space of just $(s)$.
> > >
> > >
> > > **Recurrent states** The terminology "recurrent states" that we use in the proofs of Lemma 1 (deterministic games) and Lemma 2 (stochastic games) is not completely standard:
> > > First, because we are using it for "controlled chains", whereas the standard definition is for "uncontrolled chains".
> > > Second, because the standard definition is about returning with probability 1, whereas ours is returning with non-zero probability.
> > >
> > > The standard definition of a recurrent state applies to *uncontrolled chains* and is generally defined by a state $s$ such that
> > >      $$Prob( \\exists \tau>t: t_\tau=s | s\_t = s) = 1$$
> > >
> > > Our definition for *controlled chains*, asks for the existence of a finite sequence of actions $a\_1,..., a\_m$ such that:
> > >      $$Prob( s\_\tau=s | s\_t=s, a\_t=a\_1,..., a_{\\tau-1}=a\_m)>0$$
> > > (where tau=t+m)
> > >
> > > The difference between the two definitions and your question, made us realize that it is confusing to "reuse" a common name for a new definition. A better name is probably "possibly recurrent" but we are open to suggestions.

---

> > > ### Author Response · Authors · 2025-11-28
> > >
> > > **Example of restricted fixed point** Yes, an example is probably the best:
> > >
> > > Take Tic-Tac-Toe, for which one can show that the operator defined by the right-hand-side of (8) is a strict contraction and therefore it has a unique (unrestricted) fixed point $Q^\*$. The following steps would construct a restricted fixed point $Q^\\dagger$ that is different from $Q^\*$.
> > >
> > > 1) We first pick one specific pair of saddle-point policies $(\\pi\_1^\*,\\pi\_2^\*)$.
> > >
> > > For example, we can pick a policy $\\pi\_1^\*$ for the X player that always places the 1st X on the top left corner:
> > >
> > >       X - -
> > >       - - -
> > >       - - -
> > >
> > > and pick a policy $\\pi\_2^\*$ for the O player that places the 1st O in the center slot whenever that slot is empty, and places the O in the top left corner if the 1st X was placed in the center (this latter case would never arise in response to $\\pi\_1^\*$, but could arise in response to some other policy for the X player). This policy $\\pi\_2^\*$ could lead to the following two states at time=2
> > >
> > >      - X -     or  O - -     or O in center and any other position of the 1st X
> > >      - O -         - X -
> > >      - - -         - - -
> > >
> > > It should not be too hard to accept that it is possible to find such saddle points, because each of these "opening moves" does not lead either player to a potential loss. Note that in Tic-Tac-Toe, any policy that guarantees a draw or better is a security policy and thus can be used to form a saddle-point.
> > >
> > > 2) Next, we use the saddle point $(\\pi\_1^\*,\\pi\_2^\*)$ to define the following set of pairs of policies
> > >
> > >     $$ \\Pi\^* = \\{ (\\pi_1^\*, \\pi_2) : \\pi_2\\in\\Pi_2 \\} \cup \\{ (\\pi_1, \\pi_2^\*) : \\pi_1\\in\\Pi_1 \\}$$
> > >
> > > which contains $\\pi\_1^\*$ played against all possible $\\pi_2\\in\\Pi_2$ and $\\pi_2^\*$ played against all possible $\\pi_1\\in\\Pi_1$.
> > >
> > > 3) Then compute the set of states $\mathcal{S}_{\\Pi\^\*}$ reachable under the set $\\Pi\^*$ defined in 2). This must necessarily only contain states for which there must be one mark (X or O) in the top-left corner or the center slot:
> > >
> > > When the X-player uses $\\pi\_1^\*$ we always have an X on the top-left (regardless of $\\pi_2$)
> > >
> > > When the O-player uses $\\pi\_2^\*$ we always have an O on the center slot or the top-left slot
> > >
> > > This means, e.g., that the state
> > >
> > >      - X X
> > >      - - -
> > >      - O O
> > >
> > > cannot belong to $\\mathcal{S}_{\\Pi^\*}$.
> > >
> > > 4) Take the original fixed point $Q^\*(s,a)$ and define a new function $Q^\\dagger(s,a)$ by
> > >
> > > $$ Q\^\\dagger(s,a)=Q^\*(s,a), \quad \forall s \in\\scr{S}_{\\Pi^*}, a\in\scr{A}$$
> > >
> > > $$ Q\^\\dagger(s,a)=0,\quad \forall s \not\in\\scr{S}_{\\Pi^*}, a\in\scr{A}$$
> > >
> > > This will result in a function $Q^\\dagger$ that is different from $Q^\*$ (which was the unique unrestricted saddle-point).
> > >
> > > E.g., the value of $Q\^*(s,a)$ for the example state at the end of 3) is not always zero, because this state can lead to X winning right away. But $Q^\\dagger$ will have a zero for all actions in that state.
> > >
> > > This means that $Q\^\\dagger$ is *not an unrestricted saddle-point*.
> > >
> > > 5) At this point, we basically got what we wanted because $Q\^\\dagger$ is a restricted fixed point with the $\\pi_1^\\dagger$ and $\\pi_2^\\dagger$ defined in (14).
> > >
> > > Why? Because the policies $\\pi_1^\\dagger$ and $\\pi_2^\\dagger$ will match exactly the policies $\\pi\_1^\*$ and $\\pi\_2^\*$ in step 2 over the set of states $\scr{S}_{\Pi^\\dagger}$ reachable under
> > >
> > >    $$\\Pi^\\dagger = \\{ (\\pi_1^\\dagger, \\pi_2) : \\pi_2\\in\\Pi_2 \\} \cup \\{ (\\pi_1, \\pi_2^\\dagger) : \\pi_1\\in\\Pi_1 \\}$$
> > >
> > > This is true by construction since we only changed the values of $Q^\*$ on states that could not be reached by policies on $\\Pi^\*$. In fact, by this construction:
> > >
> > > a) The set of states reachable under $\\Pi^\*$ is equal to the set of states reachable under $\\Pi^\\dagger$;
> > >
> > > b) The policies $\\pi\_1^\*$ and $\\pi\_2^\*$  will match $\\pi\_1^\\dagger$ and $\\pi\_2^\\dagger$ on the sets of states reachable by both $\\Pi^*$ and $\\Pi^\\dagger$.
> > >
> > > c) But, the policies $\\pi\_1^\*$ and $\\pi\_2^\*$  will generally *not* match $\\pi\_1^\\dagger$ and $\\pi\_2^\\dagger$ outside the sets of states reachable under $\\Pi^*$ and $\\Pi^\\dagger$.
> > >
> > > Note that c) is not a problem, because as long as one of the players plays saddle, these states will never be reached.
> > >
> > > As noted in our original response, examples of $Q^\\dagger$ restricted fixed points with a lot more default values (zeros) than the unrestricted $Q^\*$ occurred essentially in every single numerical example presented in Section 5. This is because we can see in that section that large portions of the reachable space were never visited and therefore we ended up with functions Q(s,a) that were zero (our initialization) at those states.
> > >
> > > Sorry the construction is longer than we would like, but we hope it is sufficiently specific.

---

### Official Review · Reviewer_bk2X · 2025-10-30

**Soundness:** 3
**Presentation:** 3
**Contribution:** 1
**Rating:** 2
**Confidence:** 3

**Summary:**

This paper studies two-player zero-sum turn-based Markov games. While the saddle point of this problem can be found by finding the fixed point of the Bellman operator over all state-action pairs, this paper shows that finding a fixed point over a certain subset of state-action pairs is sufficient. The paper proposes an algorithm, SPE, that finds the saddle point, possibly without exploring all state-action pairs.

**Strengths:**

The paper is nicey formatted and is easy to follow. The idea, theoretical guarantees, and the experiement are presented clearly.

**Weaknesses:**

1. The contribution of this paper does not seem strong enough for acceptance.

- Given the definition of a saddle points, specifically by Eqs. (6a) and (6b), Theorem 2 seems quite straightforward.

- The discussion is limited to deterministic games, which limits its generality.

- The theoretical sample complexity of Algorithm 1 is not provided other than that it is finite, which is limited against the compared works.

I think alpha-beta pruning is a more systematic and efficent way to implement the same idea of not searching the entire state-action space for two-player zero-sum deterministic finite-time turn Markov games that this paper focuses on. It prunes states that can only be reached by clearly suboptimal actions.
It is proved that its number of searches (which corresponds to number of sampled states in this paper) can be as small as $\mathcal{O}(\sqrt{S})$ for favorable instances, while maintaining correctness [A].
Although the setting may be slightly different, the difference seems negligible and the proposed algorithm and its theoretical guarantees seem inferior to this well-known algorithm. At least, I think some comparison must be made.

2. The proof of Theorem 2 is slightly confusing. In the proof of Theorem 2, it seems like the notations $\pi_ 1^\ast$ alternates denoting the true saddle point and the induced policy of the restricted fixed point. Lines 684-705 regard it as the induced policy from the restricted fixed point, but Line 706 states that the pair $(\pi_ 1^\ast, \pi_ 2^\ast)$ is in $\Pi^* $, which I thought was the goal of the proof. From then on it seems like $\pi_ 1^\ast$ and $\pi_ 2^\ast$ denote the true optimal policy.

[A] Knuth, D. E., & Moore, R. W. (1975). An analysis of alpha-beta pruning. Artificial intelligence, 6(4), 293-326.

**Questions:**

Theorem 2 introduces a new concept "feedback saddle point", however it is not explained in the paper.
In line 828, $s < s'$ would hold if $s'$ is reachable from $s$, but I don't think it is "if and only if".

---

> ### Author Response · Authors · 2025-11-21
> **Response to specific issues raised by reviewer bk2X**
>
> **Simplicity of Theorem 2:** *"Hindsight is 20-20" ;-)* It is worth highlighting that the most common approach to prove the "classical" result in Theorem 1 cannot be applied to Theorem 2. Specifically, it is common to show that the Q-function satisfies the fixed point condition in (8), then prove that there is a unique function that satisfies eq. 8 (typically using a contraction argument), from which it necessarily follows that the function in (8) must be the value function. Once this is done, it follows that the policies "induced" by Q are a saddle-point. This line of argument fails because the restricted fixed-point condition generally has many solutions and most of them are not equal to the Q-function. The proof of Theorem 2, instead verifies directly that the policies derived from the restricted fixed-point satisfy the saddle-point condition. The argument is not exceedingly complex (just a couple of pages in the appendix), but we believe it has merit. Please see answer below regarding the question on the proof of Theorem 2, which might have led to some misunderstanding regarding this result.
>
> **Extension to stochastic games:** Please see our general comments.
>
> **Deterministic vs stochastic:** Please see discussion in response to reviewer M6nd.
>
> **Sample Complexity of Algorithm 1:** The paper does not provide a sample complexity result because we do not believe that a *worst-case* sample-complexity result can improve upon previous algorithms. To be specific, we believe that for every exploration algorithm used to collect the samples used for the Q-function update, it is possible to construct a game for which that particular algorithm will only terminate after every single state has been explored. However, such a negative result does not mean that it is not possible to construct algorithms that visit a relatively small number of states for "interesting" games, which is the point we make with the games in Section 5.
>
> However, the reviewer's question highlights what we believe is a very important question: are there "interesting" classes of games for which we can compute non-trivial upper bounds on the number of states that need to be visited to certify policies as saddle-point. Our hope is that this work gets others interested in this question so that the community can make progress in this direction.
>
> **Alpha-beta pruning:** Alpha-beta pruning can indeed avoid a full exploration of the state/action space. However, in its classical form of a tree-search algorithm, it does not take advantage of the fact that many distinct sequences of actions (i.e., distinct tree branches) may reach the same state. Because of this, the classical form cannot solve in reasonable time even some of the smallest versions of the games we present in Section 5. For example, we have tried it on a 3x3 version of Atlatl with solve times on the order of many hours, v.s. just a few minutes for our algorithm. One could augment alpha-beta pruning with a data structure that recalls previously visited states and associated upper-lower bounds. To determine whether such a version of alpha-beta pruning is competitive or even outperforms the algorithm we proposed, one needs to specify the order by which the tree is searched as this is absolutely crucial for good performance in alpha-beta pruning. In our Atlatl tests mentioned above, we used a fixed order for the actions, which turned out to be quite bad. However, we do not exclude the possibility that, for each of the games we present in Section 5, good orders could be found. This is an interesting direction for future research.
>
> **Proof of Theorem 2:** Through the whole proof of Theorem 2 the policies \\(\pi\_1^\*\\), \\(\pi\_2^\*\\) and the set \\(\Pi^\*\\) are all induced from the function Q, which is a restricted fixed point but generally not a fixed point. To be specific, given a function Q, the set \\(\Pi^\*\\)  was defined by (14) with \\(\pi\_1^\*\\) and \\(\pi\_2^\*\\) defined by (11) for that specific function Q. However, we can see how the previous version of the paper was confusing because the reference to (11) to define \\(\pi\_1^\*\\) and \\(\pi\_2^\*\\)  seemed to imply that these policies were induced by a fixed-point and not by the restricted saddle-point. To avoid this, in the revised paper we consistently use \\(Q^\*\\) for fixed points and \\(Q^\\dagger\\) for restricted fixed points – The equation numbers also changed, but above we refer to the old equation numbers.
>
> **Questions:**
>
> The word "feedback" should not have been there: the theorem is about saddle-points as previously defined. This has been fixed.
>
> The statement with the "if and only if" was also fixed: it should have said "a total order s< s' if and only if the state s cannot be reached from  s' " An alternative (perhaps more common but lengthier) was "a total order that is a linear extension of the partial order defined by "u<v" if and only if v can be reached by u."

---

> > ### Comment · Reviewer_bk2X · 2025-11-28
> >
> > I read the authors' response and the revised paper. While the authors strongly claim that this is the first algorithm that finds certified saddle points without necessarily exploring the full state space, I think the authors are making comparisons with a wrong group of algorithms, and I would have to respecfully disagree with the claim. I believe there are algorithms that achieve the same guarantees, especially when the phrase "without necessarily exploring the full state space" is used very loosely to mean "only exploring the states that are reachable by policies considered during the learning process" for Algorithm 1. Even the article about solving checkers that the authors included in their general response shows that only a subset of states were searched, as illustrated in Fig 2.
> >
> > ---
> >
> > First, I would like to clarify why I think the works that the authors compare with (Sidford et al., 2019; Jia et al., 2019; Shah et al., 2020; Anderson et al., 2025) are not the right ones to compare. Their main challenge lies is not knowing the transition dynamics, so the focus is on providing worst-case bounds in terms of regret or sample complexity for efficiently learning unknown transition distributions. However, for the stochastic setting added in the revised version of this paper, the authors simply make an assumption that sufficient approximation is possible in finite steps without specifying how many (Assumption 2). Having no limit on the number of samples one can obtain is essentially equivalent to knowing the transition dynamics, and this is also the case for the deterministic setting since the transition becomes clear once a state-action pair is visited. Hence, I believe it is not fair to compare with the cited works.
> > In addition, one problem of not providing a sample complexity guarantee is that even when the number of visited states are small as emphasized in this paper, if the algorithm visits the same state multiple times, there is no guarantee that it will terminate early. For example, even if the algorithm visits only 5% of the states, if they are visited 1,000 times on average, then its runtime will be no better than simply running topological sort and dynamic programming.
> >
> > I list some algorithms that seem relevant. While these algorithms usually assume direct access to transition dynamics, I don't think this paper distinguishes itself from them in that aspect for the reasons above. I will mention that this is not my area of expertise and I may have misunderstood the precise meanings of the some of the works below, but I think the authors' claim is hard to justify without explaining how their algorithm differs from these groups of algorithms.

---

> > > ### Author Response · Authors · 2025-11-29
> > >
> > > **Comparison with (Sidford et al., 2019; Jia et al., 2019; Shah et al., 2020; Anderson et al., 2025)** We definitely agree with the reviewer in that the goal of our paper is very different from that of (Sidford et al., 2019; Jia et al., 2019, etc.) and, to be very specific, we do not provide complexity bounds on the number of samples needed to learn the dynamics. We cite those works as examples of turn games and/or examples of approaches to establish correctness that require a full exploration of the state space. We also cite examples of approaches that do not (e.g., Jin et al., 2022).
> > >
> > > The reviewer's comment also raises another important issue: the goal of our paper is to find saddle-point policies and *not* to learn the transition dynamics (whether deterministic or stochastic).
> > > We could have made comparison with many other papers, including basic value and policy iteration (for known transition dynamics) as well as several of the works mentioned (which we discuss in more detail below). However, all of these eventually eventually either require full-state exploration or suffer from some severe computation limitation (see comments below). In some cases, it is not hard to imagine modifying the proposed algorithms to make them more scalable, but each of these modifications would be a different paper. In the response below specific to different algorithms, we note some of these possible modifications.
> > >
> > > It is also quite correct that the checkers solver we mentioned in our response does not explore the full state-space, but it does use an independently developed checker's playing program (Chinook) to evaluate board positions and guide the search. By itself, Chinook could already defeat world champion human players. This is hardly the "general-purpose solvers" we are aiming at in this paper.
> > >
> > > **Assumption 2** There is a large body of literature on sample complexity that is devoted to figuring out precisely how many samples are needed for Assumption 2 and our Algorithm will use those bounds. These bounds are generally conservative, but we only need to apply these bounds for a 1-player game, since one of the players is always fixed at a candidate security policy.

---

> > ### Comment · Reviewer_bk2X · 2025-11-28
> >
> > - Alpha-beta pruning [A]: Alpha-beta pruning would be the one with the direct purpose of reducing the number of searched states. While my main point in the review was that alpha-beta pruning provides a more meaningful theoretical guarantees, the response from the authors only addresses the empirical performance of the algorithm, and it seems like the comparison was made with the most basic form of the algorithm. There are lots of studied heuristics for this algorithm, and I believe principal variation search with simply taking a random permutation of the actions would significantly improve the runtime while maintaining correctness. For exploiting the non-tree structure, one can simply store the lower and upper bounds of the value computed so far for each state, and there is no need for a special data structure beyond what Algorithm 1 requires for storing $Q$ values. If the board games considered in the experiment have only one reward signal of win or lose, then it suffices to store which player wins in the state once determined.
> > That said, as the purpose of the algorithm directly coincides with the purpose of this paper, I think some theoretical comparison should also be made, or at least include why the authors will not make comparisons in the discussion.
> >
> > - Best response dynamics: The proposed algorithm looks very similar to best response dynamics, where the players iteratively adopt the best response of the other players' most recent strategy. It has been extensively studied when best response dynamics succeeds [B, C, D] or fails [E]. The setting this paper considers seems like a special case considered in Proposition 1 of [B], where it shows that any two-player finite games with pure Nash equilibrium is quasi-acyclic, which implies that best response dynamics converges to the Nash equilibrium, as explained in their introduction. When viewing Algorithm 1 as best response dynamics, it is already known that it cycles or fails when the setting is slightly generalized to, for instances, simultaneous move setting (rock-scissors-paper), infinite-horizon setting [E] (see https://www.cs.tau.ac.il/~zwick/GAMES-2020/set2-tbsg.pdf for illustration), non-zero sum setting [D] (in which case it may converge to a suboptimal Nash equilibrium). The following algorithms are proved to succeed in settings where best response dynamics fails, and they still avoid full state space exploration when applied to Markov games.
> >
> > - Strategy iteration [F, G, H]: This algorithm iteratively fixes the opponents' policy and computes the best response, but here it applies one-step payoff maximization for one player and full payoff maximization for the other, and is proved to converge to the equilibrium in various settings. Naturally, the algorithm does not necessarily explore the whole state space, but only the reachable ones when one of the policy is fixed in the exact same sense with this paper.
> >
> > - Fictitious play [I, J, K] / Double Oracle [L, M]: These algorithms iteratively compute the best response considering the opponent's current and previous strategies, not only the most recent ones. They are also proved to converge to equilibria for finite zero-sum games. They are specifically designed for games with large strategy sizes, and when applied to Markov games, they would also explore only the states that are reachable by policies considered during the learning process.

---

> > > ### Author Response · Authors · 2025-11-29
> > >
> > > **alpha-beta pruning** As noted before, we have tried alpha-beta pruning with poor results. We could try different heuristics to select actions for the alpha-beta pruning tree search, in the same way that we could try different heuristics to select actions in our own algorithm. For the tests in lines 7-12 and 15-20, we used precisely the same order to explore the actions that we used for alpha-beta pruning. If our goal was to get the best possible results to date for dots-and-boxes, Atlatl, or any other of the game tests, it would make sense to explore fine-tuned strategies to explore the state-space; but we do not think that would bring significant value to this paper.
> > >
> > > In addition, and perhaps more fundamentally, it is not clear how to extend alpha-beta pruning to games for which the transition function needs to be learned by repeated game-playing. We could, as the reviewer suggested, first learn the game dynamics and then apply alpha-beta pruning, but for which portion of the state-space should we learn the dynamics? Learning the dynamics over the whole state-space would not be feasible for our large games. One possible alternative would be to combine alpha-beta pruning with some form of "on demand learning", i.e., learn the dynamics of the states as they are needed by alpha-beta pruning. That could lead to an interesting algorithm, but it would be a different paper.
> > >
> > > **Best response dynamics** In practice, it does not scale even for the most basic board games: e.g. Tic-Tac-Toe or any of the games we include in Section 5.
> > >
> > > Proposition 1 of [B] states "Any finite, two-player game with the PNEP (pure Nash equilibrium property) is quasi-acyclic."
> > >
> > > Our games have the PNEP property and are therefore quasi acyclic, which means that there is a path in policy space from any initial policy to a Nash equilibrium. The difficulty is that best-response policies are hardly ever unique (certainly not for our games) and, without further structure, it is not easy to find the best response path that ends at a Nash equilibrium. One way out of this difficulty is to construct a Markov chain in policy space that is guaranteed to explore the whole policy space: see, e.g.,
> > >
> > >     Friedman, J. W., & Mezzetti, C. (2001). Learning in games by random sampling. Journal of Economic Theory, 98(1), 55-84.
> > >
> > > However, this Markov chain will take "forever" to reach a saddle point because the number of elements of the policy space is # actions \^ # states.
> > >
> > > The difficulty we mention above is not just theoretical; instead it is very practical and we cannot get a saddle-point policy in any reasonable amount of time, even for a game as simple as Tic-Tac-Toe.
> > >
> > > **Strategy iteration** These algorithms are not scalable to large state-spaces because, when applied to turn games, they require the evaluation of the value associated with the strategy, over the whole state-space.
> > >
> > > If you look, e.g., in Algorithm 2 of [G], you see that the strategy iteration includes 3 steps:
> > > 1) start with a policy $\sigma^k$
> > > 2) compute the best counter-strategy $\tau^k$ against $\sigma^k$
> > > 3) compute the value $v^k$ associated with the pair $(\sigma^k,\tau^k)$
> > > 4) extract a policy $\sigma^{k+1}$ from $v^k$
> > >
> > > The problem is step 3 that requires computing the value $v^k$ which is defined over the whole state space.
> > >
> > > However, we conjecture that using the results in our paper, it might be possible to show that this algorithm would still converge if we compute the value $v^k$ only over a subset of the state-space. In this case, the $v^k$ would likely converge to a restricted fixed-point (as defined in our paper). Extending these results to Q-learning, we would also need assumptions like those in our paper (Assumption 1 for the deterministic case or 2-3 for the stochastic case).
> > >
> > > This should be added to our related-work section, especially [G] which is focused on turn games.

---

> > > ### Author Response · Authors · 2025-11-29
> > >
> > > **Fictitious play** Fictitious play [I,J,K] was originally defined for games without dynamics. A reasonable extension to Markov games is to associate each state with a fictitious play process. See, e.g.,
> > >
> > >       Sayin MO, Zhang K, Ozdaglar A. Fictitious play in Markov games with single controller. InProceedings of the 23rd ACM Conference on Economics and Computation 2022 Jul 12 (pp. 919-936).
> > >
> > > However, this would have complexity on the order of the number of states.
> > >
> > > **Double oracle** The double oracle algorithm maintains a growing set of policies for each player and, at each iteration, essentially finds the best response against all previous policies of the opposing player (by solving a linear program that scales with the size the number of policies tried so far). The termination condition is a saddle-point test, not unlike the one in our algorithm. If the condition is not met, then the sets of policies of the players are increased.
> > >
> > > This is a reasonable option when the policy spaces are relatively small. However, this is totally infeasible when the number of possible policies in the policy spaces are as large as # actions^ # states (with a non-trivial state-space). Note that in [L] the policy space of one of the players could grow to reasonable numbers (say 61K states, each with 32 actions), but in all examples one of the players had a very small policy space (never larger than 328). Note that the 328 was the policy space (# states ^ #action) not the number of states (which for this player was always 1).
> > >
> > > In most board games (including all we list in section 5), the policy spaces are far too large to construct the linear programs needed by double oracle policies.

---

> > ### Comment · Reviewer_bk2X · 2025-11-28
> >
> > References
> >
> > [B] Takahashi, S., & Yamamori, T. (2002). The pure Nash equilibrium property and the quasi-acyclic condition. Economics bulletin, 3(22), 1-6.
> > [C] Monderer, D., & Shapley, L. S. (1996). Potential games. Games and economic behavior, 14(1), 124-143.
> > [D] Milgrom, P., & Roberts, J. (1990). Rationalizability, learning, and equilibrium in games with strategic complementarities. Econometrica: Journal of the Econometric Society, 1255-1277.
> > [E] Condon, A. (1990). On Algorithms for Simple Stochastic Games. Advances in computational complexity theory, 13, 51-72.
> > [F] Hoffman, A. J., & Karp, R. M. (1966). On nonterminating stochastic games. Management Science, 12(5), 359-370.
> > [G] Hansen, T. D., Miltersen, P. B., & Zwick, U. (2013). Strategy iteration is strongly polynomial for 2-player turn-based stochastic games with a constant discount factor. Journal of the ACM (JACM), 60(1), 1-16.
> > [H] Auger, D., de Montjoye, X. B., & Strozecki, Y. (2021, August). A Generic Strategy Improvement Method for Simple Stochastic Games. In 46th International Symposium on Mathematical Foundations of Computer Science.
> > [I] Brown, G. W. (1951). Iterative solution of games by fictitious play. In T. C. Koopmans (Ed.), Activity Analysis of Production and Allocation (pp. 374–376). Wiley.
> > [J] Robinson, J. (1951). An iterative method of solving a game. Annals of mathematics, 54(2), 296-301.
> > [K] Berger, U. (2007). Brown's original fictitious play. Journal of Economic Theory, 135(1), 572-578.
> > [L] McMahan, H. B., Gordon, G. J., & Blum, A. (2003). Planning in the presence of cost functions controlled by an adversary. In Proceedings of the 20th International Conference on Machine Learning (ICML-03) (pp. 536-543).
> > [M] Adam, L., Horčík, R., Kasl, T., & Kroupa, T. (2021, May). Double oracle algorithm for computing equilibria in continuous games. In Proceedings of the AAAI Conference on Artificial Intelligence (35, 6, pp. 5070-5077).
> > [N] Fudenberg, D., & Levine, D. K. (1993). Self-confirming equilibrium. Econometrica: Journal of the Econometric Society, 523-545.

---

> ### Comment · Reviewer_bk2X · 2025-11-28
>
> **Simplicity of Theorem 2.** I acknowledge that I misinterpreted the theorem originally, but the revised version still doesn't surprise me. Since Theorem 2 is unrelated to the agent's knowledge of transition dynamics, the works listed above already allude to the fact that it should be true. From what I could find, it seems highly related to Corollary of Theorem 4 in [N]. This paper defines *self-confirming equilibrium* as the case where all the players' strategies and beliefs about other players' strategy satisfy two conditions:  (i) each player's current policy maximizes their reward based on their belief about other players' strategy, and (ii) the belief is correct only for the states that are visited under the current strategy. The corollary states the self-confirming equilibrium is always a Nash equilibrium for two-player games (with an additional condition of beliefs being unitary, which is always satisfied for pure strategies). The scope of the theorem is not necessarily confined to zero-sum games, which is more general than Theorem 2. While there is a distinction in the formulation of belief in [N] and restricted fixed point $Q^{\dagger}$ in this paper, the beliefs would correspond to $\arg\max_ a \mathrm{sgn}_ i(s) Q(s, a)$ which both players are aware of, and the fact that $Q^{\dagger}$ is a fixed point in $\Pi^{\dagger}$ essentially implies that both players correctly know their action values under $(\pi_ 1^{\dagger}, \pi_ 2^{\dagger})$ for the states visited by the same pair of policies, and hence they are choosing the correct action that maximizes their rewards for those states.
>
> Given these previous works, I disagree that this is a "Hindsight is 20/20" case. The current research goes far beyond what is studied in this paper. I appologize for not including these in my original review and posting them near the end of the discussion period, but I thought suggesting alpha-beta pruning for example would have been sufficient, and I was not fully familiar with the other works and needed some time to properly compare them with this paper. Given limited time, I will not ask the authors to compare their algorithm with each of the listed algorithms, but I would like to ask how the authors distinguish their algorithm and theorems in general, given these previous works.
>
> ---
>
> I have a few additional questions for the authors.
>
> 1. I am surprised that the runtime of Algorithm 1 and alpha-beta pruning differs by such a large gap, even if the latter may have been implemented without heuristics. In the experiment, how did the algorithm decide that the functions $Q_ 1$ and $Q_ 2$ have converged in lines 12 and 20 of Algorithm 1? It seems like that would require the algorithm to explore all the reachable states under fixed opponent's policy.
>
> 2. In the proof of Lemma 3, why is $F[Q](s, a) = Q^{\perp}(s, a)$ for $(s, a) \in \mathcal{Z}_ {\infty}, s \notin \mathcal{S}_ {\mathrm{recurrent}}$ (line 1049) true? I don't see any condition $Q^{\perp}$ should satisfy for those state-action pairs.
>
> ---
>
> I sincerely appreciate the authors' responses and explanations. Despite the clarifications, my concern about the novelty of this paper remains. I hope the authors will further hightlight their novelty, comparing with a larger body of literature. I would be happy to make further clarifications or explanations about my response when requested.

---

> > ### Author Response · Authors · 2025-11-29
> >
> > **Self-confirming equilibrium** We were not aware of the paper [N], but it seems very interesting. If we understand the comment correctly, the reviewer is conjecturing that our algorithm converges to a self-confirming equilibrium, which in our setting (zero-sum, unitary belief) would be a Nash equilibrium. This seems very plausible and hints that we should look at generalizations beyond zero-sum games.
> >
> > **Hindsight 20/20** We were familiar with much of the work mentioned by the reviewer (unless otherwise noted), but we still do not see previous work that can achieve what we have done.
> >
> > **Final questions**
> >
> > 1. Yes, to check for convergence in lines 12 and 20, we need to fix the policy of one of the players and then explore the full reachable state-space under that fixed policy. However, fixing the policy of one player leaves a relatively small state-space to explore.
> >
> > 2.  We apologize, there was a typo in the definition of the set $\scr{Q}^\perp$. It should include functions $Q$ that match $Q^\perp$ for pairs $(s,a)$ *outside* $\mathcal{Z}\_\infty$ not *inside*.
> >
> > Note that, in the proof of Lemma 2,  we start by establishing that $Q\^k$ converges in finite time for values of $(s,a)$ *outside* $\mathcal{Z}\_\\infty$ (see equation 56). In other words, we converge to a set of functions $\mathcal{Q}\^\perp$ defined like in (53), but with the $\forall (s,a)\in\mathcal{Z}\_\\infty$ replaced by $\forall (s,a)\not\in\mathcal{Z}\_\\infty$.
> >
> > We then use Lemma 3 to show convergence within the invariant set $\scr{Q}^\perp$ .
> >
> > With this fix, the statement the reviewer asked about should be:
> >
> > "$F$ is invariant on $\\scr{Q}\^\\perp$ simply because $F\[Q\](s,a)=Q\^\perp(s,a)$, $\forall (s,a)\\not\\in\mathcal{Z}\_\\infty$."
> >
> > This is true because $F\[Q\](s,a)$ falls in the second branch of (52)
> >
> > The typo has been fixed in the revised version of the paper.

---

### Official Review · Reviewer_f24w · 2025-10-31

**Soundness:** 3
**Presentation:** 2
**Contribution:** 2
**Rating:** 4
**Confidence:** 3

**Summary:**

This paper studies two-player zero-sum turn-based Markov games and proposes a new restricted fixed-point condition on the Q-function that guarantees saddle-point and security policies without requiring full state-space exploration. Based on this condition, the authors develop the Saddle-Point Exploration (SPE) algorithm, which alternates between best-response learning and Q-function updates. For deterministic finite games, the authors prove that SPE converges in finitely many steps to a restricted fixed point and yields a pair of saddle-point policies.

**Strengths:**

The paper introduces a theoretically grounded perspective by relaxing the full Bellman fixed-point requirement to a restricted subset of states. The resulting formulation is conceptually interesting and could inspire further work on certifiable learning under partial exploration. I didn’t check the full proof, but the finite-time convergence guarantees for deterministic games seems to be rigorous and technically solid.

**Weaknesses:**

The overall novelty appears limited. The main idea—constructing saddle-point policies from a restricted fixed-point condition—closely resembles existing work, particularly Anderson et al. (2025), which already studied finite-computation Q-learning for zero-sum turn games. The differences here mainly concern framing and sample selection, which seem incremental and insufficiently contrasted with prior work. There’s no explicit comparisons with previous works in terms of the problem settings and assumptions, theoretical results and empirical performance (note that in the Related Work section, there’s only comparison and statement for general Q-learning and zero-sum Markov games, which is not the focus of this paper.)

Further, the paper’s framing and exposition could also be improved. The introduction overstates generality by referring to Markov games broadly before restricting to turn-based settings later, which seems a bit deceiving. The contribution relative to existing literature should be articulated more precisely.

The central claim of “learning without full state exploration” is also overstated. In practice, the restricted set still corresponds to the reachable states under certain policies, which are typically the states any standard learning process would visit. This makes the contribution conceptually modest, since avoiding unreachable states is expected and does not drastically improve efficiency.

The theoretical scope is narrow, as all guarantees assume deterministic, finite-horizon, turn-based dynamics, without discussion of approximate results under stochastic or continuous settings.

**Questions:**

Could the authors clarify how the proposed “restricted fixed-point” condition fundamentally differs from the standard Bellman fixed point beyond the scope restriction? Specifically, how does it change the theoretical or algorithmic implications compared to prior formulations such as Anderson et al. (2025)?

The paper claims to learn “without full state exploration,” but the restricted set still seems to include all reachable states under the equilibrium policy. Could the authors elaborate on how to identify this restricted set and whether this reduction offers measurable computational savings?

Have the authors considered extending the theoretical results to stochastic or partially observable settings? Even a discussion of potential obstacles or approximate guarantees in such cases would help assess the broader applicability of the approach.

---

> ### Author Response · Authors · 2025-11-21
> **Response to specific issues raised by reviewer f24w**
>
> **Relationship with Anderson et al. (2025):** Both papers address the use of Q-learning iterations to "solve" zero-sum Markov turn games, and that work was indeed a significant motivation for our work, because it made the point that termination of the Q-learning iteration before exploring the full state space could lead to "arbitrarily bad" performance (when compared to a saddle-point). The authors of Anderson et al. (2025) then proceed to construct an algorithm that "protects" the Q-learning policy against a pre-defined set of policies. However, the only claim made is that the proposed algorithm guarantees a minimum reward against the given set of opponent polices; but crucially there is no statement that this guarantee is optimal (i.e., that it achieves the highest possible reward). Moreover, their numerical results specifically show that the algorithm does not reach a saddle point because there is a gap between the rewards of both players (which cannot happen for a saddle point).
>
> **Reachable set of states \& full state exploration:** We respectfully disagree in that we overstate the issue of "reachable set of states," because that is a defining feature of this work. The set of states that our iterations reach is a very small subset of the set of states that would be reached, e.g., by an epsilon-greedy exploration strategy in the full 2-player state space or almost any variations of it. Therefore it is absolutely not correct to say that the set of states visited is "typically the states any standard learning process would visit". In fact, to be able to find a saddle-point policy for our largest games (say the Atlatl game with about 200,000,000 states), it is crucial that we do *not* use an epsilon-greedy policy in line 4 of the SPE Algorithm and, instead, focus the iterations on a much narrower set of states. How much smaller this set is can be inferred from the left-hand side plot in Figure 1, where the blue and green solid lines essentially show the total number of states in the union of all sets \\(\\mathcal{S}\_{\\Pi^\*}\\) that are constructed throughout the algorithm iterations as a fraction of the total number of states that would be reachable under epsilon greedy policies. Each of these sets \\(\\mathcal{S}\_{\\Pi^\*}\\) is precisely the sets that appear in the definition of restricted fixed point and the plot shows that, as the games get larger, even the union of all these sets is still much smaller than the set of all reachable states (say about 10% for Atlatl and 5% for Dots and Boxes).
>
> We should have been clear about this, at the very least by clarifying that the percentage of states explored that we show in Figure 1 are computed with respect to the set of states that would be reached under some policy (including any common variation of an epsilon-greedy exploration). We have updated the paper to make this clear.
>
> In view of a comment from reviewer bk2X, we believe that a source for confusion may have also been the way we defined the set \\(\\Pi^\*\\) in Section 3, which seems to indicate that the set \\(\\Pi^\*\\) used to define a restricted fixed point was constructed from an actual fixed point. This is not the case: the set \\(\\Pi^\*\\) that we use to define a restricted fixed Q is constructed from the function Q itself. That is the reason why we can test whether or not a function Q is a restricted fixed point, without being able to compute an actual fixed point. We have also updated the paper to make this clear. Specifically, we now consistently use \\(Q^\*\\) for fixed points and \\(Q^\\dagger\\) for restricted fixed points.
>
> **Turn-based vs. simultaneous play:** The main reason why we restrict this work to turn-based games is that these games are guaranteed to have pure saddle-point policies. If that was not the case, the maximization in the fixed point equation (8), would have to be replaced by a max-min operation with respect to probability distributions for the selection of the players actions, e.g., like in Littman (1996). From a theoretical perspective that does not introduce significant difficulties, but for very large games the computational cost is much higher since each max-min operation requires solving a linear programming problem, rather than simply picking the largest among a finite number of values.
>
> We considered writing the paper for general zero-sum games (not necessarily turn games), but decided against it because for turn games we could aim for much larger games ( \\(>10^8\\) states) for which this approach is singularly attractive.
>
> Nevertheless, we agree with the reviewer that it is good to be clear from the start that the results are focused on turn games, so we move that statement much earlier in the introduction.
>
> **Extension to stochastic games:** To avoid further repetition, we refer to our answers above the reviewer-specific responses.
>
> **Restriction to finite time/horizon:** Please see the answers given to review M6nd.

---

### Author Response · Authors · 2025-11-21
**General comments**

# Relationship to existing work

Based on the reviewers’ comments, we feel that our manuscript under-emphasized the key contribution of our work, which is an algorithm that computes provably correct saddle-point policies for "large games" with "modest computation" in the sense that we do not need to explore every reachable state. Section 5 (numerical results) shows that we solved games with up to \\(10^8–10^9\\) states while exploring a fraction of the *reachable* states typically lower than 50%, but as low as 5% for large games. This modest computation is a defining feature of this work and translates to manageable compute times (order of hours for our largest games). The set of states that our iterations reach is a very small subset of the set of states that would be reached, e.g., by an epsilon-greedy exploration strategy (or almost any variation of it) in the full 2-player state space. Therefore it is absolutely not correct to characterize the set of states visited as "typically the states any standard learning process would visit". In fact, to find a saddle-point policy for our largest games (say the Atlatl game with about 200,000,000 states), it is crucial that we do *not* use an epsilon-greedy policy in line 4 of the SPE Algorithm and, instead, focus sampling on a much narrower set of states. Again note that we are not simply avoiding unreachable states, *we are avoiding states that would typically be explored by almost any form of exploration policy proposed in the literature*.

One can find in the literature algorithms that have been used to find strong policies for much larger games than those we deal with (e.g., Chess on the order of \\(10^{50}\\) states or Go with \\(10^{170}\\) states), but for none of these games provably correct saddle-points have been reached. A notable exception would be checkers (\\(5\times 10^{20}\\) states) with a saddle-point found in 2007 after 16 years of work (see https://www.cs.cornell.edu/courses/cs6700/2013sp/readings/06-b-Checkers-Solved-Science-2007.pdf on time needed to explore the different portions of the tree).

We can find in the literature algorithms that guarantee termination at a provably correct saddle-point after a certain minimum number of iterations. For stochastic games, "provably correct" must be understood in a probabilistic setting (typically, with probability larger than 1-delta, the algorithm terminates at an epsilon-saddle-point for arbitrarily small epsilon, delta). Our related work section lists several of these algorithms, including the ones that appear to require the smallest number of samples. Many of these papers do not provide explicit numbers for the constant "hidden" in the big-O notation for sample complexity, but the proofs of all the sample complexity results we have seen rely on full state/action exploration with high probability, so the size of the state/action joint space is a readily computable lower bound on the sample complexity. In light of this complexity, competing algorithms would require exploring at least once all \\(10^8-10^9\\) reachable states, in our examples in Section 5 while, emphatically, we can get away with exploring a small fraction of those states.

With the proposed algorithm, for every game we present in Section 5, all assumptions of our theoretical results were verified and we allowed the algorithm to run until termination, which means that we obtained provably correct saddle points; even for our largest game with about 200,000,000 states and 10-100 actions per state (Atlatl). We could not find in the literature any algorithm that could compute a *certified* saddle-point policies for games of this size, with a comparable level of computation (emphasis on "certified").

# Extension to stochastic games:

Our original paper was not quite limited to deterministic games: The definition of restricted saddle-point and the sufficient condition (Theorem 2) were both applicable to stochastic games. The SPE Algorithm was also applicable to stochastic games, but the correctness statement in Theorem 3 was indeed restricted to deterministic games. However, it is possible to prove correctness also for stochastic games, which we have done in the revised paper through a new result (Theorem 4).

In the stochastic case, termination of the loops in lines 7-12 and 15-20 (used to check whether a saddle-point was reached) must be understood in a probabilistic sense: we need convergence up to a tolerance of epsilon with probability better than 1-delta (see Assumptions 2-3). In addition, to get termination in finite time, we need to settle for an epsilon-saddle-point with some epsilon>0.

At the expense of sounding repetitive, we should emphasize that the number of samples needed in the loops 7-12 and 15-20 to get the epsilon-delta guarantee mentioned above, are generally far smaller than those required to explore every reachable state. This is because in both of these tests, the policy of one of the players is fixed.

---

### Comment · Area_Chair_Sgri · 2025-11-27
**Reminder: Please Discuss**

All Reviewers,

Thank you for your time. As the rebuttal has been available for a while, please engage in discussions with the authors and with one another. There are only a few days left before December 3.

Best,
Area Chair

---

### Meta-Review · Area_Chair_5Cqh · 2026-01-07

**Summary:**

Overall, the discussion focused on questions of novelty and significance. The authors and the reviewers engaged in a productive discussion that rectified some misunderstandings. Significant amount of effort was put into the responses. However, by the end of the process there were still concerns on the side of the reviewers regarding the strength of the contribution and framing within the literature. My recommendation reflects this consensus.

**Reviewer Concerns:**

While the authors successfully addressed several questions from the reviewers, the main outstanding concerns center on novelty and contribution.

Reviewer bk2X had concerns about the representation of the literature, and how the paper fits within it. The reviewer proposed a long list of papers, and the authors discussed how their paper differed. This seemed to help improve the scope of the paper, but unfortunately it remains unclear whether the reviewer would agree with the characterization of the authors.

Reviewer f24w also mentioned some concerns about the scoping of the generality of the paper.

Reviewer M6nd had some questions that were quickly clarified. They also seemed convinced the contribution is somewhat modest. For one, they mentioned the lack of quantitative bounds on the number of visited states to be a weakness of the paper, though the authors mention leaving that as a direction of future research.

The latter sentiment about the lack of upper bounds is found also in Reviewer PmtW's review, who seemed unhappy about the lack of theoretical backing to the claim that the author's method is better than prior ones.

Overall, there are recurring concerns about the contribution. The rebuttal was responsive but the reviewers did not seem to be swayed.

**Reviewer Scores:**

None of the reivewers mentioned a desire to update their score. Those reviewers that participated in multiple rounds of discussion with the authors explicitly mentioned at time not being swayed by the arguments. My prediction is that all reviewers would have maintained their score.

---

### Decision · Program_Chairs · 2026-01-26

Reject